# Measurement report: Greenhouse gas profiles and age of air from the 2021 HEMERA-TWIN balloon launch

Tanja J. Schuck[1], Johannes Degen[1], Timo Keber[1], Katharina Meixner[1], Thomas Wagenhäuser[1], Mélanie Ghysels[2], Georges Durry[2], Nadir Amarouche[3], Alessandro Zanchetta[4], Steven van Heuven[4], Huilin Chen[4,7], Johannes C. Laube[5], Sophie L. Baartman[6,*], Carina van der Veen[6], Maria Elena Popa[6], and Andreas Engel[1]

[1]Institute for Atmospheric and Environmental Sciences, University of Frankfurt, Frankfurt, Germany
[2]Groupe de Spectrométrie Moléculaire et Atmosphérique, Université de Reims, France
[3]Institut National des Sciences de l'Univers Division Technique, Meudon, France
[4]Centre for Isotope Research, University of Groningen, The Netherlands
[5]Institute of Climate and Energy Systems: Stratosphere (ICE-4), Jülich Research Centre, Germany
[6]Institute for Marine and Atmospheric research Utrecht, Utrecht University, The Netherlands
[7]The School of Atmospheric Sciences, Nanjing University, China
[*]now at Meteorology and Air Quality Group, Wageningen University and Research Centre, The Netherlands

**Correspondence:** Tanja J. Schuck (schuck@iau.uni-frankfurt.de)

**Abstract.**

Within the HEMERA balloon infrastructure project, a stratospheric balloon carrying a multi-instrument payload to a maximum altitude of 31.2 km was launched on 12th August 2021. Aboard the openly constructed gondola, several types of instruments were used for simultaneous air sampling and in-flight measurements to characterize climate relevant trace gases in the stratosphere and in the troposphere, and to compare and evaluate different instrumental approaches and sampling techniques. For observations of the main greenhouse gases carbon dioxide ($CO_2$), methane ($CH_4$), nitrous oxide ($N_2O$) and sulfur hexafluoride ($SF_6$), sampling with AirCores, flask sampling and in-flight spectrometry were deployed. Overall, results from different methods agree well. While better precision is achieved for the post-flight measurements of AirCores and flask sampling, in-situ spectrometry provides a higher degree of detail on the vertical structure of the $CH_4$ profile. Age of air was derived from mixing ratios of $CO_2$ and $SF_6$. As seen in previous studies, higher values were obtained from $SF_6$ than from $CO_2$. Correcting for chemical losses, maximum values of 4.4–5.1 years were derived from $SF_6$ mixing ratios at altitudes above 20 km compared to 4.2–5.0 years from $CO_2$ mixing ratios. The resulting dataset should be well suited for multi-tracer approaches to derive age of air, in particular in combination with a large suite of halocarbons measured from flask samples and one more AirCore which are reported by a companion publication.

## 1 Introduction

High-altitude balloons continue to be the only means for in-situ observations of chemical composition at altitudes that cannot be reached by aircraft, i. e. above ca. 20 km. Lightweight instrumentation, such as for example ozone sondes, can be lifted with small weather balloons, that can be launched routinely. Many trace gases, however, can only be measured with more complex

instruments or from sampled air analysed post-flight in the laboratory (Ehhalt, 1980; Fabian, 1981). Cryogenic air sampling is an established method for the efficient collection of air samples in the stratosphere (Ehhalt, 1974; Lueb et al., 1975; Schmidt et al., 1984, 1987) to obtain observations of trace gas profiles from the stratosphere.

Such data are for example relevant to constrain potential changes in stratospheric circulation induced by climate change (Austin and Li, 2006; Engel et al., 2009; Stiller et al., 2012; Eichinger et al., 2019; Abalos et al., 2021). It is also of interest to perform measurements of ozone depleting gases directly at the altitudes where ozone depletion occurs (Ray et al., 2002; Brinckmann et al., 2012; Krysztofiak et al., 2023). For substances which are measurable with remote-sensing methods, data from balloon-borne air samples can also be used for satellite retrieval validation or to validate ground-based measurement networks such as the Total Carbon Column Observing Network (TCCON) and the Network for the Detection of Atmospheric Composition Change (NDACC) deploying Fourier transform infrared spectrometers (Zhou et al., 2018). This was for example recently demonstrated for vertical profiles of HCFC-22 (Chlorodifluoromethane, $CHClF_2$) measured by the ACE-FTS satellite instrument (Kolonjari et al., 2024) using data from the analysis of flasks sampled at ground-based sites, on board aircraft and sampled during four earlier flights of the identical cryo sampler used for the HEMERA TWIN launch. However, due to the high costs and safety aspects of such launches, profiles from large high-altitude balloons remain sparse in their spatial and temporal coverage (Krysztofiak et al., 2023; Ray et al., 2024).

Over the last decade, air sampling with AirCores, based on an idea initially proposed by Tans (2009), has been established as a method for measurements of vertical profiles of $CO_2$ and $CH_4$ (Karion et al., 2010; Membrive et al., 2017; Engel et al., 2017; Wagenhäuser et al., 2021). AirCores are a lightweight air sampling tool based on stainless steel tubes that are open at one end and closed at the other. Making use of the pressure changes with altitude, air is passively sampled into the tube during the descent from high altitudes to the ground. They are often sufficiently lightweight to be carried by small weather balloons. This approach is complementary to classical flask sample collection at distinct altitudes. While the latter averages over few well-defined sampling intervals, AirCores provide a continuous profile. AirCore samples need to be analysed quickly after sampling to minimize the averaging effects of molecular diffusion within the sample tube and to achieve the best possible vertical resolution. Also, sub-sampling of AirCores for later analysis using discontinuous analytical methods - at the expense of losing altitude resolution - is possible (Laube et al., 2020). Sub-sampled AirCore samples, as original flask samples, can be analysed for many species even after longer times of storage, depending on the chemical stability of compounds in the flasks. The 2021 launch of the HEMERA-TWIN gondola described in this measurement report allowed the simultaneous deployment of several AirCore packages for inter-comparison of different AirCores and for comparison with other measurement methods, which is not possible with small weather balloons because of their payload weight restrictions.

To investigate stratospheric transport time scales, the concept of mean age of air has proven to be a useful tool. As the stratospheric circulation cannot be observed directly, a quantity that can be derived from observations of trace gas mixing ratios is needed to characterize stratospheric transport. Mean age of air can be used to diagnose the current overall strength of the large-scale Brewer-Dobson but also allows to investigate changes that might be a consequence of climate change (Hall and Plumb, 1994; Waugh and Hall, 2002; Engel et al., 2009; Garny et al., 2024b).

Commonly, the mean age of air is interpreted as the mean transit time that it took for all contributions to an observed air parcel to arrive at the observation location from their respective entry points into the stratosphere. The calculation relies on a reference time series of mixing ratios measured at a reference surface. Often, the tropical tropopause is chosen as the reference surface. For the lower stratosphere of the mid latitudes, more sophisticated approaches take into account cross-tropopause transport in the extra-tropics as well (Hauck et al., 2020; Wagenhäuser et al., 2023; Ray et al., 2024). Generally, the stratospheric age of air can be calculated from observations of long-lived trace gases which have a monotonous trend in the troposphere, for example $SF_6$ or, when de-seasonalised, $CO_2$ (Engel et al., 2002; Bönisch et al., 2009; Ray et al., 2014; Engel et al., 2017; Garny et al., 2024a; Ray et al., 2024). Also halocarbons have been used to derive age of air (Daniel et al., 1996; Volk et al., 1997; Harnisch et al., 1999; Leedham Elvidge et al., 2018). Age of air values need to be corrected for the acceleration of tropospheric trends, as trace gas mixing ratios in general show non-linear trends (Plumb and Ko, 1992; Volk et al., 1997; Engel et al., 2002). In addition, the chemical sinks of $SF_6$ may introduce significant biases to the calculation (Stiller et al., 2012; Ray et al., 2017; Kovács et al., 2017; Leedham Elvidge et al., 2018). Recently, Garny et al. (2024a) proposed a model-based correction scheme to account for chemical sinks of $SF_6$. For $CO_2$, also the seasonal cycle in the troposphere needs to be taken into account at least in the lower stratosphere (Bönisch et al., 2009; Andrews et al., 2001; Diallo et al., 2017).

Here, we report on the HEMERA-TWIN balloon launch of 2021 which aimed at sampling and analysing stratospheric air to measure atmospheric trace gases using four types of instruments: in-situ spectrometric analysis, cryogenic air sample collection in stainless steel canisters, bag sampling, and air sampling by means of AirCores. This combination of instruments allows to compare vertical profiles of the long-lived greenhouse gases $CO_2$, $CH_4$, $SF_6$ and $N_2O$ measured at different altitude resolution and of the mean age of air derived from $SF_6$ and $CO_2$. As discussed in our companion paper, the collection of air samples provides additional data on mixing ratios of halocarbons, many of them being strong greenhouse gases and ozone depleting substances (Laube et al., 2025).

## 2 Methodology

### 2.1 The HEMERA-TWIN balloon launch

HEMERA is a balloon infrastructure project offering balloon flights for research and innovation. It is funded by the European Commission within the Horizon 2020 program and is coordinated by the French space agency CNES (Centre National d'Etudes Spatiales). In August 2021, the openly constructed TWIN gondola shown in Fig. 1 was used as part of the HEMERA flight programme to carry a suite of instruments to measure the vertical distribution of several long lived greenhouse gases and ozone-depleting substances. The TWIN gondola with its name referring to the symmetric frame structure, has been used before in similar studies and is considered to be suited for whole air sampling without contamination of the sampled air (Engel and Schmidt, 1994). The open structure avoids contact of the sampled air with surfaces and thus reduces the probability of contamination.

The launch took place on 12th August 2021 at 21:18 UTC from the European Space and Sounding Rocket Range (Esrange) in Kiruna, Sweden located 68° N, 21° E at approximately 330 masl altitude. Ascent took place over approximately 3:45 hours

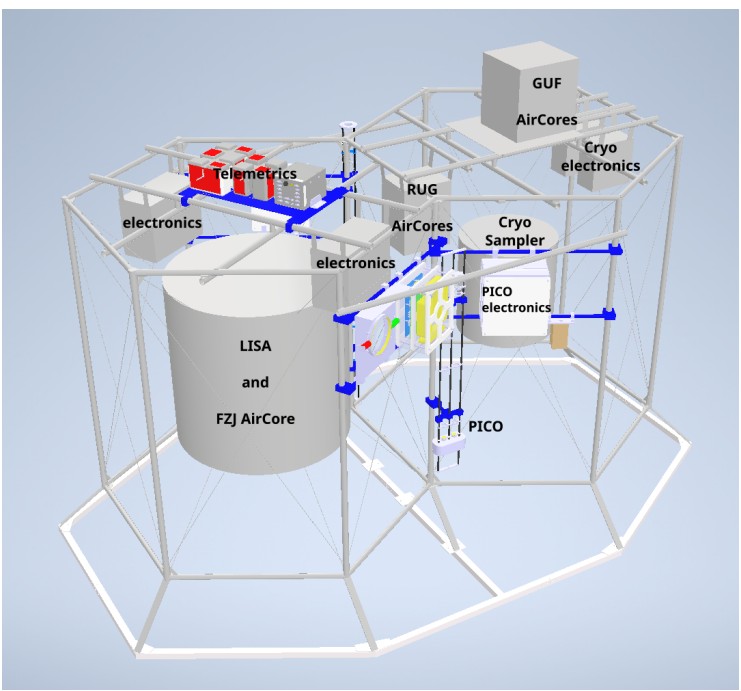

**Figure 1.** Layout of the TWIN gondola for the HEMERA 2021 launch. The gondola height is 2.2 m, footprint is 2.6 m x 4 m, and the total weight including the instrumentation amounts to approximately 345 kg. Its name refers to the symmetric structure of the gondola frame.

at an average altitude rate of 4.5 m/s and the balloon reached a maximum altitude of 31.2 km around 23:12 UTC, where it spent approximately 7 min, before descent started. The valve-controlled slow descent was over approximately 3:50 hours with an average vertical speed of -1.5 m/s between 31 and 13 km and an average vertical speed of -8.2 m/s below 13 km after separation of the gondola from the main balloon. Touch-down was at 3:02 UTC. All sampling of air was during the descent to avoid

possible contamination of sampled air by the balloon or the gondola itself. The height of the tropopause was derived from radiosonde data during the balloon ascent by calculating the slope of the temperature profile. The tropopause height has been assigned at 10.5km where the temperature change with altitude first reached the value of 2K/km following the WMO definition.

The individual instruments of the payload are listed in Table 1, and Fig. 1 shows the layout of the payload within the gondola structure. The gondola is 2.2 m in height over a base area of 2.6 m x 4 m. In summary, the balloon carried the cryogenic whole

air sampler (Schmidt et al., 1987), five different AirCores (Engel et al., 2017; Wagenhäuser et al., 2021; Laube et al., 2025) and two mid-infrared diode laser spectrometers, Pico-SDLA $CH_4$ and Pico-STRAT Bi Gaz, for in-situ measurements of $CH_4$ and $CO_2$ (Ghysels et al., 2011, 2014).

Position data as well as ambient pressure and temperature were recorded by the Pico-STRAT Bi Gaz instrument. The position data include GPS altitude, latitude and longitude from a GNSS system, pressure measurements are performed with

a ParoScientific Inc. absolute gauge, and ambient temperature values are recorded by three fast-response temperature sensors (Sippican).

**Table 1.** The payload of the HEMERA-TWIN gondola on 12th August 2021. AirCores and air samples were analysed post-flight with Quantum Cascade Laser Spectroscopy (QCLS), Cavity Ring Down Spectroscopy (CRDS) and Gas Chromatography (GC) coupled with an electron capture detector (ECD) or a mass spectrometer (MS), whereas Pico-SDLA $CH_4$ and Pico-STRAT Bi Gaz are in-situ instruments measuring in flight.

| instrument | analysis |
|---|---|
| Cryogenic Whole Air Sampler | offline GC-MS, GC-ECD, CRDS |
| AirCores | offline CRDS, QCLS, sub-sampling for GC-MS |
| Lightweight stratospheric air sampler LISA | offline QCLS and GC-MS |
| Pico-SDLA $CH_4$ | mid-infrared in-situ spectrometry of $CH_4$ |
| Pico-STRAT Bi Gaz | mid-infrared in-situ spectrometry of $CO_2$ and $H_2O$ |

**Table 2.** Uncertainties of Pico-SDLA $CH_4$ measurements obtained in-flight as a function of pressure level.

| pressure range | single spectrum precision | | 1 s precision | |
|---|---|---|---|---|
| [hPa] | [ppb] | [%] | [ppb] | [%] |
| < 50 | 20 | 8.7 | 9 | 8.6 |
| 50–100 | 20 | 3.3 | 9 | 3.4 |
| 100–250 | 15 | 2.9 | 6 | 2.9 |
| 250–60 | 57 | 1.7 | 25 | 2.1 |
| > 600 | 48 | 2.6 | 20 | 1.4 |

In addition, the payload included a newly constructed air sampler for the collection of large air samples in foil bags which was developed based on the LIghtweight Stratospheric Air sampler (LISA) (Hooghiem et al., 2018). This sampler will be described in a separate publication and data are not included here The total weight of the payload was approximately 345 kg.

In the post-flight analyses of air samples and AirCores, $CO_2$ mixing ratios were measured on the WMO X2019 scale (Hall et al., 2021), $CH_4$ on the WMO X2004A scale (Dlugokencky et al., 2005) and $N_2O$ on the WMO X2006A (Hall et al., 2007). $SF_6$ mixing ratios are reported on the WMO X2014 scale (NOAA, 2014).

## 2.2 The Pico-SDLA spectrometers

Pico is a balloon-borne spectrometer developed to probe vertical profiles of atmospheric $CH_4$ and $CO_2$ (Ghysels et al., 2011, 2014). During the HEMERA-TWIN flight, two Pico instruments were launched: Pico-SDLA $CH_4$ and Pico-STRAT Bi Gaz ($H_2O$/$CO_2$). The Pico-STRAT Bi Gaz ("Bi Gaz": dual gas in French) spectrometer is an evolution of the former Pico-SDLA instruments to suite long duration observations. This adaptation has been initiated within the STRATeole 2 balloon-borne project. (Carbone et al., 2024). The Pico-SDLA $CH_4$ instrument performed well, whereas Pico-STRAT Bi Gaz measurements

suffered from undesired electromagnetic interference for which the source remains undetermined, resulting in spectrum defor-
mations for $CO_2$. Therefore, $CO_2$ measurements are unusable for this flight.

Pico-SDLA $CH_4$ deploys a mid-infrared distributed-feedback laser emitting at 3.24 µm. The laser beam is propagated in
the open atmosphere over a total absorption path length of ~3.6 m after multiple reflection. The total weight of the device
is approximately 8.5 kg. The simple and robust design of the optical cell minimizes mechanical vibrations, thereby limiting
variations of the spectra baseline. Pico-SDLA $CH_4$ was integrated into the gondola in a vertical position. The slow ascent and
descent reduced mechanical vibrations, thereby increasing the optical cell instrumental stability.

The wavelength of the laser emission is tuned by ramping the laser driving current every 10 ms. Atmospheric mixing ratios
are retrieved from the in situ absorption spectra using a molecular model in conjunction with in-situ atmospheric pressure and
temperature measurements. Ambient pressure is measured by an absolute pressure transducer with 0.01 % accuracy (ParoSci-
entific Inc.), measurements are averaged over 0.5 s. Ambient temperature is measured using three fast-response temperature
sensors (Sippican) with an uncertainty of 0.2°C and a resolution of 0.1°C on the temperature reading. Measurements are av-
eraged over 1 ms with outliers removed. Sensors are located at each end of the optical cell and at its center. The sensors are
known to be susceptible to solar and infrared radiation, but no correction was necessary as measurements took place during
night. The temperature uncertainty was improved by an inter-comparison program (Oakley et al., 2011).

The noise of one spectrum is about $4 \cdot 10^{-4}$ in absorption units. Using single spectra, the measurement precision scales
from 48 ppb at ground, down to 15 ppb around the tropopause. For a 1 s averaging time, the precision varies from 25 ppb
in the troposphere down to 6 ppb in the UTLS (cf. Table 2). Spectroscopic laboratory work has been conducted in order to
determine the appropriate molecular model, accounting for temperature-related effects (Ghysels et al., 2014). This improved
the measurement accuracy. The uncertainty budget includes the uncertainty due to the frequency axis and baseline interpolation,
the uncertainty due to experimental noise and spectroscopy as well as the uncertainties of pressure and temperature. Table 2
lists the measurement uncertainties of Pico-SDLA $CH_4$ from the ground up to the balloon ceiling.

## 2.3 Vertical profile measurements with AirCore

The gondola carried three different AirCore packages, from Goethe University Frankfurt (GUF), University of Groningen
(Rijksuniversiteit Groningen, RUG) and from Forschungszentrum Jülich (FZJ). An overview is given in Table 3.

The AirCore sampling system is based on a concept first presented by Karion et al. (2010) from an idea originally developed
and patented by Tans (2009). AirCores consist of long and narrow stainless steel tubing which at launch time is closed at one
end and open at the other. Prior to launch the AirCore is filled with a gas of well-known composition. It evacuates due to
decreasing ambient pressure during ascent and reversely samples ambient air with increasing pressure during descent. To
avoid loss of sample air or contamination, AirCores may be equipped with a mechanism to automatically close the tube
upon landing. After recovery, the sample is analysed for trace gas mixing ratios with a continuous-flow gas analyser, and the
resulting measurements are attributed to the sampling altitudes. Altitude attribution was based on pressure readings from the
Pico-SDLA $CH_4$ instrument for both the GUF and RUG AirCores. To attribute the measured trace gas mixing ratios to sampling
altitude, the pressure- and temperature-dependent amount of sampled air is calculated as a function of altitude and related to

**Table 3.** Properties of the five different AirCores in the payload.

| Name | outer diameters | wall thickness | length | inlet | comment |
|---|---|---|---|---|---|
| AirCore GUF003 | 8, 4, 2 mm | 0.2, 0.2 , 0.12 | 20, 40, 40 m | with $Mg(ClO_4)_2$ dryer automatically closed | |
| AirCore GUF005 | 8, 4, 2 mm | 0.2, 0.2, 0.12 | 20, 40, 40 m | with $Mg(ClO_4)_2$ dryer automatically closed | spiking experiment failed data not used |
| RUG AirCore wet | 3/16", 1/8" | 0.01", 0.005" | 37, 39 m | automatic closing failed | contamination in $CO_2$ between 11-14 km |
| RUG AirCore dry | 3/16", 1/8" | 0.01", 0.005" | 36, 38 m | with $Mg(ClO_4)_2$ dryer closing failed | contamination in $CO_2$ between 11-14 km |
| FZJ AirCore | 1/4", 1/2" | 0.25 , 0.5 mm | 170, 60 m | with $Mg(ClO_4)_2$ dryer | see (Laube et al., 2025) |

the amount of sample air measured at a constant flow as a function of measurement time. A small amount of fill gas remains in the AirCore tube, that during descent is pushed towards the closed end of the AirCore. During analysis, which is performed in the reverse direction, the remaining fill gas marks the start of the AirCore sample in the measurement time series. In this procedure an easily distinguishable fill gas facilitates the analysis.

Including electronics, AirCores from GUF and RUG each add only 3 kg to the payload which in a single instrument package makes them deployable with small weather balloons. Deploying them as part of a large instrument package allows the comparison of different configurations. Another larger AirCore, developed and operated by FZJ, was sub-sampled for laboratory GC-MS analysis of halogenated tracers (Laube et al., 2020). Results thereof are discussed jointly with the GC-MS results from the cryogenic whole air sampler in a companion paper (Laube et al., 2025).

For GUF, the main scientific objective of AirCore measurements is the determination of the mean age of air from $CO_2$. Therefore, the AirCores are geometrically designed such that the highest vertical resolution is obtained for the stratosphere (Membrive et al., 2017). They are composed of three different sections with smaller diameters towards the stratospheric end to reduce mixing due to diffusion during the time between sampling and measurement (inner diameters: 7.6, 3.6, 1.76 mm; outer diameters: 8, 4, 2 mm; length: 20, 40, 40 m). Further details have been described by Engel et al. (2017) and Wagenhäuser et al. (2021). The AirCores are constructed from custom-made stainless steel tubing which has been silanised, as suggested by Karion et al. (2010), using Silconert2000® to reduce wall effects and to enhance sample stability during storage. Both AirCores were equipped with $Mg(ClO_4)_2$ dryers at the inlet and were automatically closed upon landing.

For the 2021 TWIN gondola launch, one GUF AirCore was equipped with a CO spiking experiment as described by Wagenhäuser et al. (2021) to test the altitude attribution. Because the spiking experiment failed, results of only the AirCore with default configuration are presented here. The initial fill gas of both GUF AirCores had $CH_4$ and $CO_2$ mixing ratios close to those expected in the middle stratosphere but was spiked with CO, resulting in a CO mixing ratio of 1436.41 ppb. Mixing with

the remaining fill gas is taken into account during the retrieval as described by Wagenhäuser et al. (2021). Thus, the uppermost part of the AirCore profile can be used for scientific evaluation as well.

Starting ~3 hours after landing, GUF AirCores were analysed for CO, $CH_4$, $CO_2$ and $H_2O$ using a Picarro G2401 CRDS, and results are reported as dry mixing ratios. The measurement data are calibrated in two steps. First, the raw data was processed with instrument specific parameters that are valid over the long term. Therefore, a linear calibration curve for each component using analyser-specific slope and offset values was applied. These device characteristics were determined in laboratory experiments prior to the campaign. Secondly, the values were corrected for instrumental drift with a day-specific offset determined by measuring a calibration gas tank immediately after the AirCore analysis.

Altitude attribution was performed as described by Engel et al. (2017) and Wagenhäuser et al. (2021). The start and end points of the AirCore sample in the measurement time series were determined using the known mixing ratios of the remaining fill gas that the AirCore tube is filled with prior to the launch and the push gas that is used to push the sample air towards the Picarro instrument during the post-flight analysis. The vertical resolution of the GUF AirCores ranges from about 1000 m at 25 km to better than 300 m around the tropopause and in the troposphere. However, the geometry of the AirCore plays a central role in this uncertainty, so the three individual sections of the GUF AirCore with their different internal diameters and lengths must be taken into account. At lower altitudes the effect of molecular diffusion on the vertical resolution is larger because of the wider tube diameter. At the top of the profile, mixing in the analyzer cell during post-flight analysis is the dominating effect. Further details of the AirCore data analysis including the altitude attribution and fill gas correction were reported by Wagenhäuser et al. (2021).

The RUG AirCores are similarly designed with smaller diameters towards the stratospheric end to reduce mixing during sampling and sample recovery, each consisting of two sections of different diameters: outer diameter 3/16" and 1/8" with wall thickness ~0.01" and ~0.005", lengths: 37 m and 39 m for one AirCore, 36 m and 38 m for the other. The sections were connected with an externally glued union. One AirCore's inlet was equipped with a $Mg(ClO_4)_2$ dryer, while the other AirCore's inlet was left open, to investigate possible water effects on the retrieved profiles. After landing and retrieval, both AirCores were measured on a dual-laser Aerodyne QCLS, detailed below in subsection 2.5 and in Vinković et al. (2022); Tong et al. (2023). The altitude attribution was realized following the approach described in Membrive et al. (2017).

Unfortunately, in both AirCores, the glue connector caused a contamination issue for $CO_2$ between 11-14 km of altitude. The affected data in this range are not reported, and are visible as gaps in the profiles. Upon landing, the closing mechanism of both RUG AirCores malfunctioned, likely due to prolonged cold soak during the flight. The closing attempt drained the batteries, shutting down the data loggers and no temperatures were recorded. Warming of the open-ended tubing between landing (03:02 UTC; T ~-45°C) and capping by the recovery team (approximately 04:15 UTC; T unknown) will have led to a loss of sample from the RUG AirCores, which in consequence will have led to a too-low altitude attribution of the profile. The exact loss and attribution bias cannot be stated with certainty as no temperature data for a volumetric correction were recorded. Indicatively, heating by 10°C would lead to a low bias in altitude attribution of ~300 m in the lower troposphere while the stratospheric part of the profile would be less affected.

Additional uncertainty exists for the stratospheric measurements and altitude attribution: the top and bottom of the retrieved profiles are biased by mixing with the remaining fill gas and the gas employed to push the AirCore air in the instrument. Given uncertainties regarding the gas composition and the actual mixing fractions of the profiles and fill/push gases over the analysis time, the upper and lower parts of the RUG profiles are not reported.

## 2.4 Cryogenic whole air sampling

The cryogenic whole air sampler holds 15 stainless steel sample flasks with volumes 0.58 L (five flasks) or 0.31 L (10 flasks) which were evacuated before the launch. Each flask has an individual inlet system, made of metal and glass, which is opened and closed interactively through telemetry commands from ground. Inlet lines open towards the bottom of the sampler to avoid contact of the sampled air with any equipment. During sampling, ambient air is cryogenically trapped with liquid neon. The air sampler contains a 10 L reservoir of liquid neon which is filled prior to launch. Further details of the sampler were described by Schmidt et al. (1987). During the HEMERA 2021 launch, the five larger sample flasks were equipped with cotton filters to scrub ozone for accurate measurements of carbonyl sulfide (COS) replacing manganese dioxide on glass wool previously used for this purpose (Andreae et al., 1985; Hofmann et al., 1992; Persson and Leck, 1994; Engel and Schmidt, 1994).

In total, 14 samples were successfully collected during descent at altitudes ranging from 30.8 km up to 13.5 km, final sample pressures ranged from 8 bar to 33 bar, corresponding to a total sample volume of 4.6–19.1 L STP. The sample time varied between 43 s and 731 s, corresponding to an altitude range of 85 m to 1278 m depending on the respective descent velocity. Different types of post-flight analyses of the sampled air were performed at laboratories at GUF, RUG and FZJ.

## 2.5 Air sample analysis

All flasks of the cryo sampler were analysed at GUF with GC-MS for halocarbons, with gas chromatography/electron capture detection (GC-ECD) for CFC-12 and $SF_6$, and with the continuous-flow CRDS used for AirCore analysis for $CO_2$, $CH_4$ and CO. The sample volume used for these analyses was two times 1 L for GC-MS, and approximately 0.27 L and 0.11 L for GC-ECD and CRDS analysis, respectively. The cryo sampler was then transferred to RUG, where the continuous-flow QCLS used for AirCore analysis was deployed to measure $CO_2$, $CH_4$ and $N_2O$ from the sample flasks. This used approximately 0.25 L of sample volume. Last, samples were analysed for $SF_6$ at FZJ using a GC-MS setup that needs 0.25 L of sample volume.

At GUF, all flask samples were analysed for halogenated compounds with a GC-MS set-up almost identical to the one described by Hoker et al. (2015) and Schuck et al. (2018), but deploying a quadrupole mas spectrometer in selected ion monitoring mode only. In addition, the air samples were analysed with a semi-continuous GC-ECD set-up for CFC-12 and $SF_6$ (Engel et al., 2006; Jesswein et al., 2021; Wagenhäuser et al., 2023) and by high resolution CRDS deploying the instrument described in section 2.3 for analysis of $CO_2$, $CH_4$ and CO. Because CO is known to grow in the stainless steel canisters (Novelli et al., 1992), only $CO_2$ and $CH_4$ data are presented. $SF_6$ data from the GC-ECD instrument are measured on the SIO-05 scale, and a conversion factor of 1.0049±0.002 was applied to convert to the WMO X2014 scale (Prinn et al., 2018). One sample, collected without cotton scrubber at 19.3 km altitude, was excluded from further analysis for $CO_2$ and $CH_4$ due to an unrealistically high mixing ratio of COS above 5 ppb as detected during GC-MS analysis and a $CO_2$ mixing ratio above

420 ppm. These high values might indicate a stability issue during storage. Mixing ratios of $SF_6$ and $N_2O$ are shown, as these compounds are chemically inert and unlikely to be affected by storage effects. Furthermore, a sample equipped with cotton scrubber collected at 17.9 km altitude was excluded because unrealistically high mixing ratios of several trace gases were measured, including $CO_2$, $SF_6$ and several halogenated compounds, which points to a contamination of this sample. All measurements of halogenated tracers with the GC-MS setup from the cryogenic air samples will be presented in a companion paper (Laube et al., 2025).

At RUG, in September of 2021, the cryosamples were analysed on a quantum cascade laser spectrometer (QCLS; model TILDAS Dual, Aerodyne Research Inc., MA, USA). Its first laser (scan centered around wavenumber 1275.5) observes $CH_4$, $N_2O$ and $H_2O$, while its second laser (around wavenumber 2050.6) observes COS, $CO_2$ and CO. The cavity of the analyser is maintained at a pressure of $50 \pm 0.002$ Torr (~66 hPa) and a temperature of $250 \pm 0.002$°C. Under these conditions the equivalent volume of the optical cavity is ~10 cm$^3$ (geometric volume ~150 cm$^3$), and a precision better than 0.6 ppb, 0.12 ppb, and 0.20 ppm is attained for $CH_4$, $N_2O$ and $CO_2$. respectively (1$\sigma$ of individual samples, collected at 1 Hz). Sample flowrate is ~50 sccm.

The instrument stabilization is of 'double exponential' character, i. e., exhibiting an initially rapid approach to the final value, but then taking a long time to become stable. This depends on species and is more pronounced for $CO_2$ than for $CH_4$. The measurement duration of ~5 minutes, chosen to conserve precious sample, does not for all samples result in quantitative replacement of the previous sample. In order to maximize the value of our analysis, a double exponential function was fit to the measurement data and used to predict the true sample value, i. e. the asymptote of the function). The deviations between the asymptote and the mean of the last 60 seconds of a sample was for almost all samples smaller than the uncertainty of either. That means that this method does not make unjustified assumptions and does not add significant uncertainty to the results.

Measurements are calibrated against multiple compressed air working standards (prepared in-house). Each working standard was measured repeatedly before, during and after the samples to control for conceivable drift. QCLS response functions were obtained by linearly fitting the measurements of the standards to their assigned values, after linearly interpolating these measurements in time. The obtained time-dependent response functions were applied to raw measurements, and the curve fitting procedure is repeated to obtain the final sample results. Running in tightly controlled laboratory conditions, the performance of the QCLS was excellent and the uncertainties in our final values are dominated by the inaccuracies in the assigned values of our working standards (i.e., not instrumental noise or drift), taken to be $\pm 1$ ppb, $\pm 0.3$ ppb, $\pm 0.10$ ppm, respectively for $CH_4$, $N_2O$ and $CO_2$. We note, however, that this assessment may not hold true for stratospheric samples, of which the mixing ratios of multiple trace gas species are significantly lower. For these samples, unknown (but unexpected) non-linearities of the QCLS response may reduce the attained accuracy. Such conceivable but unlikely bias cannot be compensated for due to absence of suitable calibration gases and data were evaluated assuming a linear response of the instrument.

The analytical procedure at FZJ consists of three main steps: 1) cryogenic extraction and pre-concentration of trace gases at ~-78°C, immediately followed by thermal desorption at ~95°C, 2) separation by gas chromatography (Agilent 6890 GC with a 60 m GS GasPro column and a temperature program from -10 to 200°C), and 3) detection with a triple-sector mass

**Table 4.** Instrumental precision and average error of analyses of flask samples and AirCores.

| | CRDS | QCLS | GC-ECD | GC-MS |
|---|---|---|---|---|
| used for | GUF Cryo<br>GUF AirCore | RUG Cryo<br>RUG AirCore | GUF Cryo | FZJ Cryo |
| $CO_2$ | 0.01 % (0.025 ppm) | 0.05 % (0.2 ppm) | – | – |
| $CH_4$ | 0.05 % (0.2 ppb) | 0.03 % (0.6 ppb) | – | – |
| $SF_6$ | – | – | 0.6 % (0.05 ppt) | 0.6 % (0.06 ppt) |
| $N_2O$ | – | 0.03 % (0.12 ppb) | – | – |

spectrometer (Waters AutoSpec MS) in selected ion monitoring mode. Further details are described in the companion paper by Laube et al. (2025). $SF_6$ mixing ratios are reported on the WMO X2014 scale with an average precision of 0.6 % (0.06 ppt).

Precision values for the analysis of AirCores and cryo sampled flasks are summarized in Table 4. All instruments meet the minimum requirements to ensure that data are useful as defined for the World Meteorological Organisation Global Climate Observing System programme (WMO, 2024). These are 0.5 ppm for $CO_2$, 5 ppb for $CH_4$ and 0.3 ppb for $N_2O$. As ideal requirements, beyond which no further improvement seems necessary, 0.1 ppm, 1 ppb and 0.05 ppb are defined for these three gases. This ideal data quality goal is meet for $CO_2$ by the CRDS instrument and the QCL instrument comes close. WMO does not define a data quality target for $SF_6$.

## 2.6 Age of air calculations

The mean transit time that it took for all contributions to an observed air parcel to arrive at the observation location from their respective entry points into the stratosphere is defined as the mean age of air. Thus, the observed mixing ratio of an inert trace gas at some place in the stratosphere is determined through the distribution of transit times, called the age spectrum, and the long-term change in its mixing ratio at the entry point. Commonly, the age spectrum is mathematically described as an inverse Gaussian function, and the mean age of air is the first moment of this distribution (Hall and Plumb, 1994; Waugh and Hall, 2002). Mean age of air values were derived from $SF_6$ measurements and independently from simultaneous $CO_2$ and $CH_4$ measurements following the procedure described by Garny et al. (2024b) using the *AoA_from_convolution* python package version 1.0.0 (Wagenhäuser et al., 2024). In brief, this method calculates the expected mixing ratios of an inert trace gas through a mathematical convolution of the age spectra for different mean age values and the mixing ratios time series at the entry point. The mean age is then determined from the best match between the observations and the mixing ratios resulting from this forward calculation. The derived mean age is the age value for which the convolution best fits the observed mixing ratio.

To calculate mean age from tracer observations, also the time series of the respective tracer at a reference surface is needed. For this purpose, the *AoA_from_convolution* package uses the NOAA Greenhouse Gas Marine Boundary Layer Reference for $SF_6$, $CO_2$ and $CH_4$ trace gas mixing ratio time series at the tropical surface $\pm$ 17.5°around the equator (Lan et al., 2021; Garny

et al., 2024c). The inverse Gaussian describing the age spectrum is parameterised assuming a ratio of moments as described
in Garny et al. (2024b). Mean age values below 1 year are omitted due to numerical reasons of the software implementation. Regarding SF6, these mean age calculations do not account for the mesospheric sink, which leads to apparently older $SF_6$ mean ages (Leedham Elvidge et al., 2018; Garny et al., 2024a).

For $CO_2$, the software first uses $CH_4$ mixing ratios to account for stratospheric $CO_2$ production from $CH_4$ degradation. This corrected $CO_2$ mixing ratio is then used to derive the mean age. Note that the seasonal cycle of $CO_2$ propagates into the lower
stratosphere, and it is impossible to disentangle the seasonal cycle from the long-term increase. Therefore, mean age values below 2 years derived from $CO_2$ measurements are problematic (Garny et al., 2024a).

## 3   Comparison of trace gas mixing ratio and age of air profiles

Figures 2 and 3 show the vertical profiles of $CH_4$ and $CO_2$, $N_2O$ and $SF_6$ respectively. For $CH_4$, also data from the Pico-SDLA $CH_4$ instrument are included, labelled short as "PICO". It is the only instrument that also provides data for the balloon
ascent. For $CO_2$ and $N_2O$, only data from AirCores and the cryo samples are shown, $SF_6$ was only analysed from the flask samples. Left hand panels for each gas show the vertical profile of absolute mixing ratios, right hand panels show mixing ratios relative to results from the analysis of air samples at GUF, except for $N_2O$ which was not measured at this laboratory. High resolution measurements have been averaged over the sampling period of each individual sample. The error bars in the right hand panels indicate the variability of the high resolution data over the sample collection time.
In the troposphere, $CO_2$ mixing ratios show variability between 403 ppm and 413 ppm, and $CH_4$ mixing ratios vary between 1920 ppb and 1980 ppb in the two AirCore datasets. The high resolution Pico-SDLA chCH4 data exhibits a tropospheric mixing ratio range from 1920 pbb to 2100 ppb. Above 10 km, $CH_4$ mixing ratios decrease slowly up to 17 km and decrease steeper with altitude above. Above 20 km, several layers of low $CH_4$ mixing ratios are apparent in the Pico-SDLA $CH_4$ data which cannot be resolved by the other measurement methods. $CO_2$, in contrast, starts to increase at an altitude of approximately 7 km and
starts to decrease at around 14 km altitude. A minimum is reached around 24 km altitude. Comparing results from analysis of AirCore and flask samples, $N_2O$ behaves similar to $CH_4$.

Comparing the two AirCore systems (i. e., GUF and RUG), for which data analysis and altitude attribution is done independently, there is an offset of approximately 200–700 m with the RUG AirCores being attributed systematically to lower altitudes. The main difference in the processing procedures from GUF and RUG is the fill gas correction. For the RUG AirCores, no fill
gas correction is performed, therefore profiles do not extend as high as GUF profiles. An additional uncertainty is introduced for RUG AirCores because of the failure of the automatic closing of the inlet after touch down. This may have cause a loss of sample that could not be corrected.

Comparing the two profiles retrieved from the RUG AirCores with and without drying on sampling, the water vapour did not cause a major bias in the retrieved $CO_2$ and $CH_4$ molar fractions. In fact, the biggest differences were found in the stratospheric
part of the profile, where the atmospheric $H_2O$ content is negligible. In the troposphere, differences between the two AirCores at comparable altitudes reached up to 0.4 ppm and 2 ppb for $CO_2$ and $CH_4$, respectively. However, the stratospheric part of the

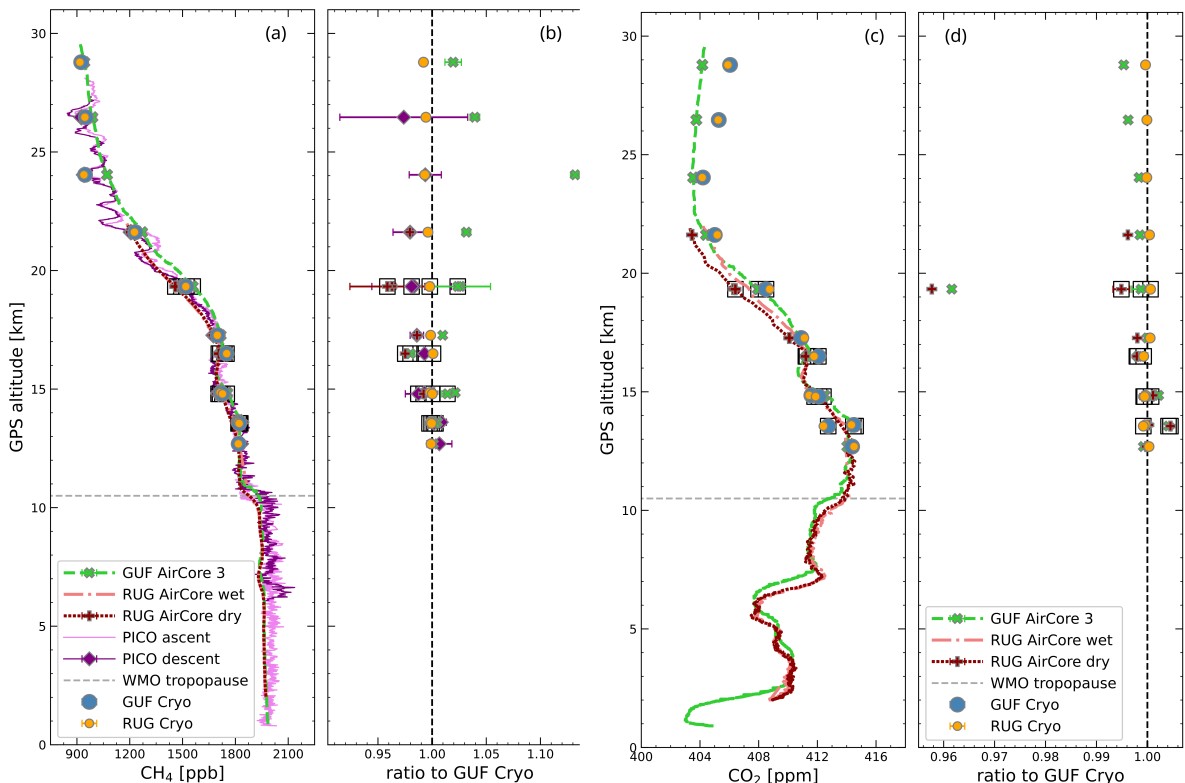

**Figure 2.** Vertical profiles of $CH_4$ (a) and $CO_2$ (c) and comparison with results from the cryo samples as measured at GUF (b and d). To compare results of the high resolution observations with the samples, the mean value over the sampling period has been calculated using the standard deviation as error bars. Error bars may be smaller than symbol size. Mixing ratios of samples with the cotton scrubber are highlighted by the square symbol.

profiles showed differences up to 1.3 ppm for $CO_2$ and 5 ppb for $CH_4$. The reasons behind these differences remained overall unclear, but we speculate they could be ascribed to some remaining mixing with the AirCores' fill gases, or to the interaction of the $Mg(ClO_4)_2$ dryer with other gas species in the stratosphere.

All flask samples were analysed post-flight at GUF and RUG with the identical instruments that were used for post-flight analyses of AirCores. For both laboratories, good agreement within the respective instrumental precisions is found. For $CO_2$, the average difference is 0.14 ppm, varying between 0.04 ppm and 0.22 ppm, for $CH_4$ it is 4.3 ppb, varying between 1.8 ppb and 7.4 ppb. Because of the good agreement between the two datasets, in the following, measurement results obtained from the cryo sampler at GUF are used as reference for comparison except for $N_2O$, which was only measured at RUG. Around

14.8 km and 13.5 km altitude overlapping samples with and without a scrubber were collected, although for technical reasons

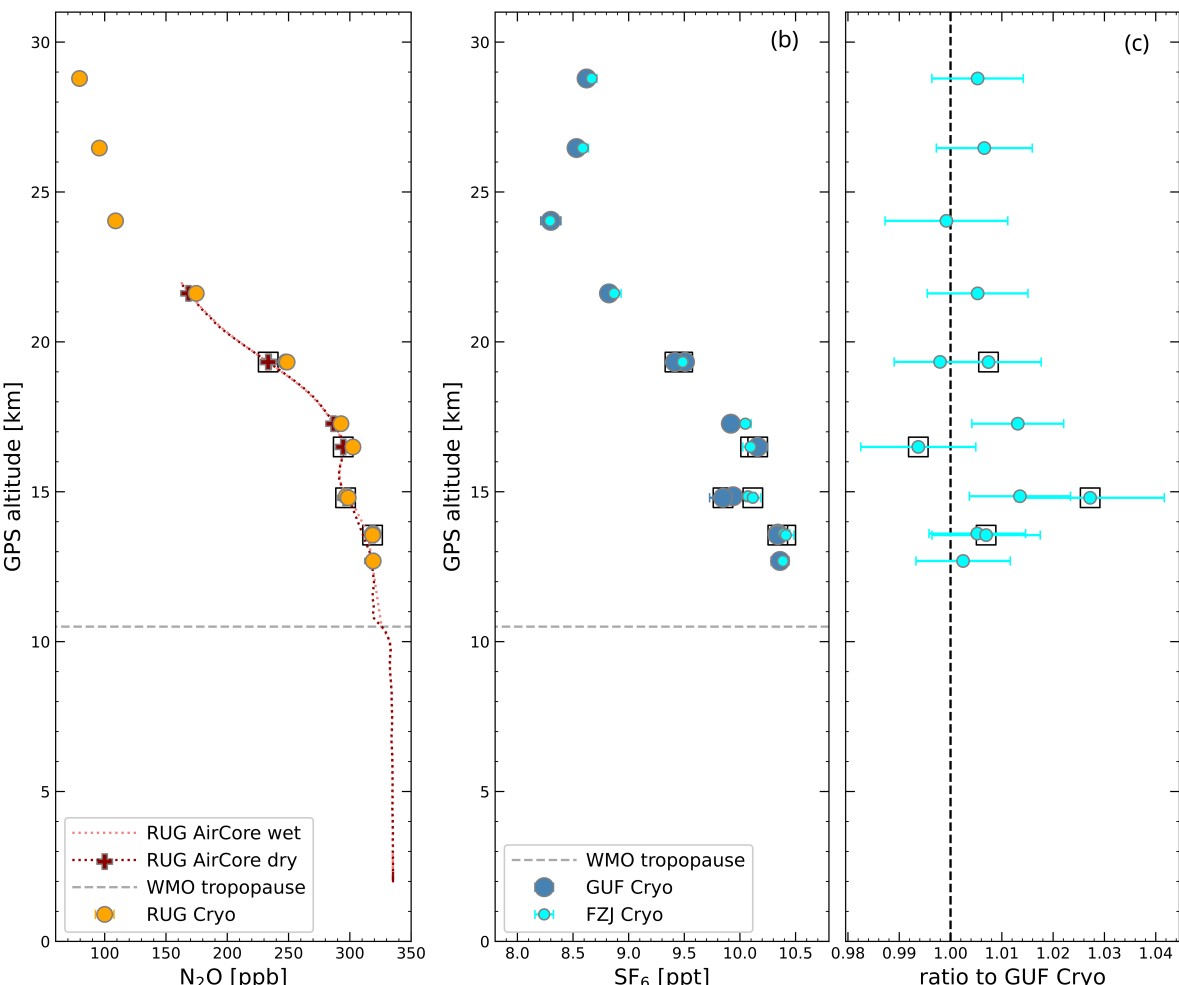

**Figure 3.** As Fig. 2, for $N_2O$ and $SF_6$. $N_2O$ was only measured at RUG and no inter-laboratory comparison is possible. Vertical profiles of $SF_6$ analysed from air samples with GC-ECD (GUF) and GC-MS (FZJ). Error bars of absolute values are smaller than symbol size. For better differentiation AirCore profiles are plotted as dashed or dotted lines, this does not imply a coarser resolutions as profiles are continuous.

not covering the exact same altitude range, as only one sample flask could be opened or closed at a time. Mixing ratios of $CO_2$, $CH_4$ and $N_2O$ agree well for those two sample pairs.

The Pico-SDLA $CH_4$ is the only instrument of the payload that provides data for the balloon ascent. Although for the lowest altitude part of the profiles the time difference between ascent and descent is almost 7 hours, the two measurements
agree closely. The spectrometer can resolve small structures in the stratosphere much better than the AirCores which provide a smoothed profile in comparison to Pico-SDLA. When averaging over sample collection times of the cryo sampler, which are

between 43 s and 731 s, very good agreement is found with $CH_4$ mixing ratios measured post-flight in the laboratories at GUF and RUG. Also, for the sample collected at 24 km altitude, when the spectrometer recorded a local minimum of $CH_4$ mixing ratios which is not captured by the AirCore observations, both independent post-flight analyses agree. On panels (a) and (c) of Fig. 2, the variability of the Pico-SDLA $CH_4$ data is reflected by the larger error bars of the integrated Pico-SDLA $CH_4$ data, most pronounced for the air sample collected at 26.6 km. On average, integrated Pico-SDLA $CH_4$ data deviate from the sample analysis results in Frankfurt by 9 ppb, with a minimum deviation of 6 ppb and a maximum difference of 25 ppb. In the troposphere, Pico-SDLA $CH_4$ data are noisier than in the stratosphere, and they are slightly offset towards higher mixing ratios compared to the AirCore profiles. Mixing ratios follow those derived from AirCore measurements, but with more fine structure, as the Pico-SDLA $CH_4$ directly records in situ data, whereas AirCores are a technique with an inherent averaging kernel. Above 20 km, the Pico-SDLA $CH_4$ profile reveals several layers of lower $CH_4$ mixing ratios, which cannot be resolved by air samples nor the AirCores.

AirCore data have also been averaged over the sampling interval of each cryo sampler flask. Differences relative to the direct measurement of $CO_2$ and $CH_4$ might partly arise from the uncertainty in altitude attribution. In addition, AirCores do provide a continuously sampled profile, but due to molecular diffusion some averaging and smoothing with altitude occurs. This becomes most evident comparing the $CH_4$ profile derived from AirCore measurements to the in situ recording of $CH_4$ mixing ratios from the Pico-SDLA $CH_4$ spectrometer. The Pico-SDLA $CH_4$ recorded several fine structures above 20 km altitude which are not apparent in the AirCore profiles. Compared to the cryo sampler analyses GUF, GUF AirCore $CH_4$ integrals tend to be higher on average by 35 ppb, with a minimum difference of 2.2 ppb and the maximum difference of 125.0 ppb, which is observed for the sample collected during a $CH_4$ minimum of the Pico-SDLA $CH_4$ profile.

In $CO_2$, AirCore profiles tend to be at lower mixing ratios in comparison to the cryo sampler analyses, in particular at higher altitudes. For data from GUF, the maximum deviation is 1.55 ppm, best agreement found is within 0.14 ppm. Similar results are obtained for the AirCores from RUG for $CO_2$ and $N_2O$.

Figure 3 (b) and (c) compares the results of $SF_6$ measurements from the air samples in two different laboratories, at GUF and at FZJ. In Frankfurt, $SF_6$ is measured with GC-ECD, in Jülich with GC-MS. Similar to $CO_2$, $SF_6$ mixing ratios decrease above the tropopause with the steepest vertical gradient occurring between 17 and 24 km. Results of the two independent measurements agree within their respective uncertainty with a mean difference of 0.04 ppt, varying from 0.01 ppt to 0.27 ppt, corresponding to a relative difference range 0.08 % to 2.7 %. For each instrument, results for the overlapping samples with and without the ozone scrubber agree within the uncertainty. As for $CO_2$ and $CH_4$, the steepest gradient in the $SF_6$ mixing ratios occurs between 17 and 24 km altitude and, as shown in Fig. 4, there is a clear correlation between $SF_6$ and the two other greenhouse gases.

Figure 5 compares age of air values derived according to Garny et al. (2024b) from mixing ratios of $CO_2$ and $SF_6$ with the tropical marine boundary layer mixing ratios as reference time series (Wagenhäuser et al., 2024). Comparing the results from the flask samples to the AirCore analysis exemplary for the GUF AirCore, the systematically higher $CO_2$ mixing ratios obtained from the flask samples at the highest altitudes are reflected as lower age of air vales. However, the general structure

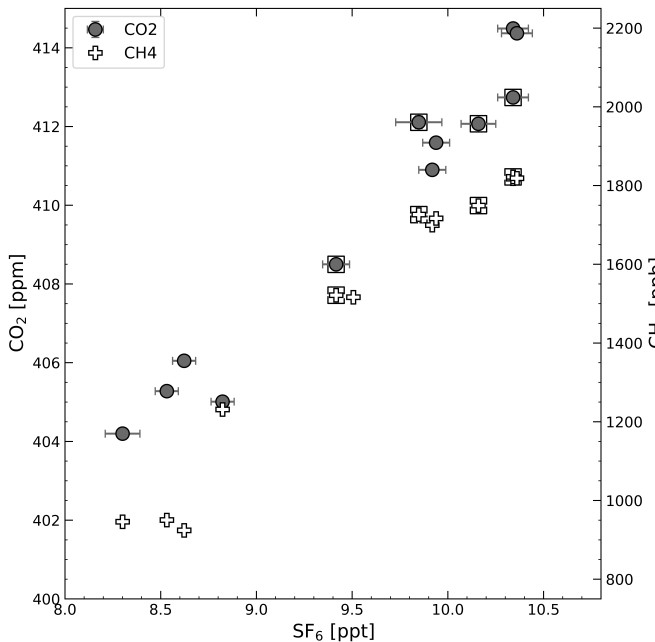

**Figure 4.** Correlation of $CO_2$ and $CH_4$ with $SF_6$ as analysed from air samples. Mixing ratios of samples with the cotton scrubber are highlighted by the square symbol.

of the profiles agree, in particular at altitudes below 20 km. This confirms that AirCores are a measurement technique suited to derive age of air profiles from $CO_2$ observations.

For $CO_2$, systematically lower ages are derived, with the difference increasing with altitude. This agrees with findings by Ray et al. (2024), who consistently analysed a large number of dataset from aircraft and balloon borne measurements for the periods 1994–2000 and 2021–2024, including AirCore data for the latter period. In the 2021 HEMERA TWIN dataset, the difference increases to 1.5 years at altitudes around 24 km and above. Using the bias correction described by Garny et al. (2024a) to account for chemical sinks of $SF_6$ reduces the $SF_6$ age by 0.05 years to 1.67 years with an average reduction of 0.55 years. The corrected values range from 1–5 years with maximum ages of 4.4–5.1 years above 20 km altitude. The corrected $SF_6$ agrees better with the value derived from $CO_2$ with the average difference reducing from 0.66 years to 0.08 years when applying the correction.

## 4 Conclusions

Within the HEMERA balloon infrastructure project coordinated by the French space agency CNES, the openly constructed TWIN gondola was launched from Kiruna, Sweden, in August 2021. The gondola was equipped with two different air samplers, three different types of AirCores and the mid-infrared diode laser spectrometer Pico-SDLA $CH_4$ for in-situ measurements

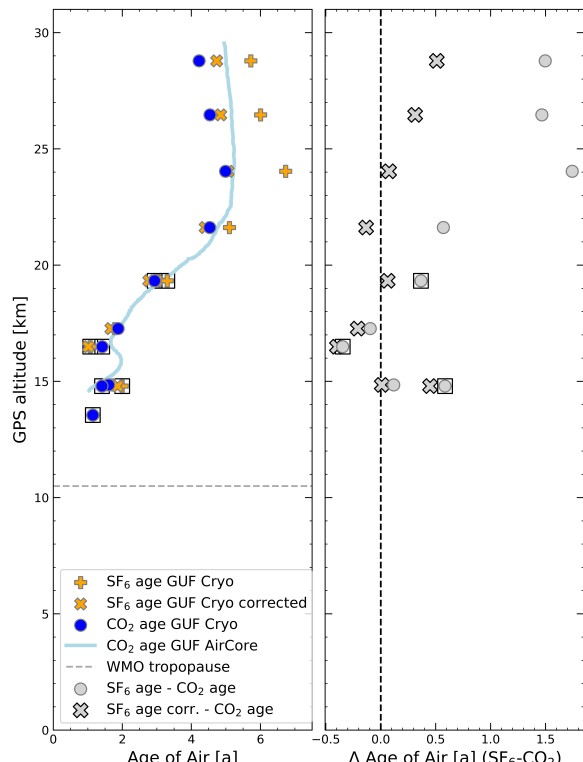

**Figure 5.** Age of air values derived from $CO_2$ and $SF_6$ mixing ratios of flask air samples and AirCore $CO_2$ mixing ratios. Samples with the cotton scrubber are highlighted by the square symbol. "X" markers represent Age of air values from $SF_6$ mixing ratios corrected using the method by Garny et al. (2024a).

of $CH_4$. A maximum altitude of 31.2 km was reached. Here, we reported on analysis results for $CO_2$, $CH_4$ and $SF_6$ from cryogenically collected air samples in stainless steel flasks. Samples were analysed post-flight in different laboratories using optical and gas chromatography based measurement techniques. Mixing ratios of all three greenhouse gases agree well for the different analyses.

Results are compared to vertical profiles of $CO_2$ and $CH_4$ derived from AirCores and for $CH_4$ to the measurements with the
395 Pico-SDLA. The latter instrument records more fine structure at altitudes above 20 km, which is not apparent in the AirCore data with much smoother profiles. While the agreement between the Pico-SDLA $CH_4$ measurements and the air sample analysis is very good for $CH_4$, AirCore derived $CH_4$ deviates, likely because the AirCore cannot resolve the observed minima in the $CH_4$ profile. Additional uncertainty arises from the altitude attribution of the AirCore profiles. This becomes more apparent in the case of $CO_2$, for which the difference between air sample analyses and AirCore profiles seem to increase with altitude. In
addition, differences between two independent AirCore datasets are observed, with altitude differences of individual features of up to 300 m.

For $CO_2$ and $SF_6$ age of air was derived from the observations following the approach by Garny et al. (2024b). At altitudes of 24 km and above, maximum ages between 5.8 and 6.5 years were obtained from $SF_6$ mixing ratios which reduced to 4.4–5.1 years after correction of the chemical sink according to Garny et al. (2024a). Age of air values derived from $CO_2$ are systematically lower, with the difference increasing with altitude, in agreement with finding from other datasets. Up to an altitude of 25 km age of air derived from AirCore analysis of $CO_2$ agrees well with flask sample data, at higher altitudes differences occur. Accounting for chemical sinks of $SF_6$, the $SF_6$-based age of air decreases and better agrees with $CO_2$-derived age of air within 0.5 years. Recently, Ray et al. (2024) proposed a new technique of deriving age of air from simultaneous measurements of several long-lived trace gases. The data set presented here, containing $CO_2$, $CH_4$ and $SF_6$, should be well suited for this approach. The dataset, which will be complemented by further data of halogenated long-lived trace gases (Laube et al., 2025), will enable further age of air evaluations.

*Data availability.* Observational data are available from https://zenodo.org/records/13918431 with doi: 10.5281/zenodo.13918431.

*Author contributions.* AE, AZ, GD, JL, MEP, NA, SLB, SvH, TJS, TK and TW operated the instrumentation in the field and contributed to data analysis, CvdV contributed to the campaign preparation and laboratory work. AZ, JL, SvH, SLB, MEP, TJS and TW performed post-flight sample analysis. JD and KM performed further data analysis of GUF AirCore data. TJS drafted the manuscript. All coauthors contributed to the scientific discussion and improvements of the manuscript.

*Competing interests.* TJS and AE are members of the editorial board of *Atmospheric Chemistry and Physics*.

*Acknowledgements.* We are grateful for the support from the technical staff before, during, and after the campaign, especially from Anne Richter, Andreas Sitnikov, Jochen Barthel, and Vicheith Tan (FZJ) and Laurin Merkel (GUF). We acknowledge the support from local staff at ESRANGE and from the CNES team. We acknowledge NOAA Global Monitoring Laboratory for providing surface measurements of $CO_2$ and $SF_6$ for derivation of age of air.

*Financial support.* This research has been supported by the DFG collaborative research program "The Tropopause Region in a Changing Atmosphere" TRR 301 – Project-ID 428312742. and by the European Research Council (grant no. EXC3ITE (678904) and COS-OCS (742798) and HEMERA (730790)).

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
