# Peer review of "Measurement report: Greenhouse gas profiles and age of air from the 2021 HEMERA-TWIN balloon launch"

_EGUsphere, 2024_

## Referee Comment (RC2)

**Review of "Measurement report: Greenhouse gas profiles and age of air from the 2021 HEMERA-TWIN balloon launch" by Schuck et al.**

**General Comments:**

The manuscript outlines measurements made during the August 2021 large-balloon-assisted launch of the HEMERA-TWIN gondola, which hosted discrete flask sampling systems, AirCore samplers and continuous in situ instrumentation for measurement of long-lived atmospheric trace gases. Profile data are compared to flask sampler data, and stratospheric mean age of air is calculated from flask and AirCore samples of $CO_2$ and $SF_6$.

Overall, this measurement report provides updated large-balloon observations from Kiruna and new and important observation-based calculations of stratospheric mean age of air worthwhile of publication. Some scientific, technical and organizational comments below are recommended to be addressed before publication, however (see specific comments below).

- There are several references to measurements made, but not used in this manuscript and/or described elsewhere (Laube et al., 2024 – not currently accessible). Text throughout the manuscript could be tightened up a bit to not cause undue confusion.
- More details are needed in the onboard sampling systems and analytical descriptions, in addition to the mean age calculation in order for methods to be sufficiently understandable and repeatable by readers.
- The visualization and description of the profiles in each figure could be improved upon, and separate AirCore profile data figures could be merged into one multi-panel figure for comparisons of species' profiles and their corresponding description within the text.

**Specific Comments:**

Figure 1: I am wondering where each package was located on the gondola. Could some packages in this figure be somehow labeled to indicate where measurements were made?

Table 1:

- Five – L82? (or three – L118?) different AirCores were used here, but it would be nice to have these (with perhaps IDs for each AirCore and dimensions, configuration, dryer, no dryer, etc.) described in Table 1 explicitly. Please also add ambient pressure and temperature sensors to Table 1 as these are equally important measurements.
- LISA and the FZJ AirCores are included in this table, but data are not presented. This is contrary to the approach taken in Table 3. Pico-STRAT Bi GAz is also not added in Table 1. I would suggest either adding all, but with "*" for measurement artifacts and/or descriptions in other, cited papers, or remove otherwise.

L125: Not all AirCores (e.g. FZJ) were analyzed by continuous-flow analyzers. Please correct/clarify. Were similar fill gas mixtures used for each AirCore?

L139/L159: It seems that the GUF and RUG AirCores samples were processed somewhat differently, from fill/push gas mixing corrections to the altitude attribution of the samples.

Please describe in more detail what the differences are between both GUF and RUG methods so that this can be assessed better.

Section 2.3: You might consider a 2.3.1 'GUF AirCores' vs. 2.3.2 'RUG AirCores' for organization

L169-L172: There is no mention of an ambient water vapor measurement onboard the gondola. It is difficult to examine water effects on the profiles (either dried or undried) without this type of measurement. For example, if the atmosphere is dry, no effects will be seen, so some clarification here and/or below in the description of differences seen between these profiles should mention something about the average tropospheric water vapor content above Kiruna.

Please define $Mg(ClO_4)_2$. Please also describe in more detail the altitude attribution used here. Is this similar to Tans, 2022 or Wagenhauser et al., 2021? How do these methods compare/differ?

Table 3:
   - The text mentions that this table describes precision for both flask and AirCore measurements (L240), so the caption should mention this.
   - It is a bit confusing with some species mentioned in this table. COS is mentioned for the RUG QCLS, but Table 3 precisions and results are not described. FZJ analysis precision by GC-MS is stated in Table 3 for only SF6 results with other species presented elsewhere. Please see comment above for Table 1.

Section 2.5: Could you please clarify the progression of air sample analysis and how many flasks were analyzed by each organization? Are all flask samples analyzed for halogenated species at GUF by GC-MS, by GC-ECD for CFC-12 and SF6, and by CRDS for CO2, CH4 and CO and then subsequently transferred to RUG and FZJ for additional species analysis?

Section 2.6: Please describe in a bit more detail how *AoA_from_convolution* works for the reader within the text. It might also be clarified in subsequent text that mean ages are calculated and used.

L271: Combining CO2, CH4, N2O profiles in a single figure would be advantageous to your description of the profiles within the text, and would allow the reader to better compare/contrast the variability between profiles.

Figures 2-6:
   - It is very difficult to distinguish the colors between RUG AirCore 'wet' and 'dry'.
   - Cryosample and other markers overlap each other at times and it's difficult to see what's underneath of these. Consider decreasing marker size here.
   - Line width on PICO data covers AirCore profile data as well; consider reformatting so that all profiles can be sufficiently visualized.

L274: The altitude registration offset looks to be greater than 300 m (maybe 500-800m?), hence my request for additional description of the altitude attribution techniques above and potentially additional explanation for this offset in L281.

L303: The AirCore averaging kernel or diffusion is stated multiple times throughout the text. Could you include calculated uncertainties for this either in the text or in the figures above to indicate how much of an uncertainty the AirCore storage diffusion has?

L314: Can you offer any thoughts as to why AirCore $CO_2$ (and $N_2O$) seems to be lower than the cryosampler data of the same species? Could it be a problem with one or either of these sampling methods?

**Technical Comments:**
L17: 20 km 'MSL'?
L21: It is perhaps important to note that balloon data are also needed to validate and "calibrate" remote-sensing instrumentation like those in the TCCON and NDACC, not only to supplement them.
L24: What type of balloon platform (large or small-volume balloon) and instrument (flask, in situ) was used here? Please describe, as AirCore could potentially be used as a low-cost option that refutes the statement in L25.
L29: Please define chemical species names of $CO_2$ and $CH_4$, but also elsewhere in the manuscript for completeness.
L33: The AirCore approach, providing higher resolution profiles and additional air sample measurements, seems like an improved method over discrete flask sampling methods. Consider replacing the word "complementary"
L39: What does HEMERA-TWIN stand for, if anything?
L40: What are 'reference methods'? It might be more accurate to say that the gondola instrumentation allowed simultaneous comparison of several different measurement methods and not to say that one is more accurate than the other.
L42: Please describe why calculation of the mean age of air is significant and useful in a bit more detail for context.
L49: 'monotonic'
L59: ..bag sampling, [and] air sampling by means of …
L75: an average 'ascent' rate … and 31.2km 'MSL'?
L82: Please define BONBON acronym
L86: Please define LISA
L94: In Pico-STRAT Bi GAz, what does STRAT, Bi and GAz signify?
L133: Here and throughout the manuscript, text switches from spelling out 'Frankfurt', or 'University of Frankfurt' and 'Groningen' vs. using the acronyms for each organization. Consider using just acronyms for simplicity.
L149: part of the AirCore [$CO_2$ and $CH_4$?] profiles can be used…
L153: these are reported as dry mole fractions
L171: in subsection [space] 2.5 (Vincović et al. 2022; Tong et al, 2023).
L174: ..and [are] visible?

L185: upper and lower parts, or just upper parts?

L290: Above states that data is only collected upon balloon descent, so please correct.

L293: Can you please state over how long of a sample collection time this averaging takes place?

L308: The fine structure here is not seen in AirCores due to the diffusion of the samples, presumably.

L309: Delete "University of Frankfurt,", "by", and add on average, [by]

L326: age of air 'values'

L335: value derived 'from'

Figure 7: The x-axis label should be years, correct? Caption: "..markers represent 'Age' of air"

L372: Does the NOAA product come with a citation reference?

---

## Author Comment (AC1)

**Schuck et al., The interhemispheric gradient of SF6 in the upper troposphere**

**Response to comments by Anonymous Referee #1**

We thank the anonymous reviewer for the thorough reading of the manuscript and their suggested improvements. All comments are addressed in the following with the reviewer's comments printed in blue, and the responses in black.

**General comment**

I believe that this work has been achieved on the basis of the long history of experimental advances in balloon observations by the European scientific community. I suppose that earlier works such as Schmidt et al. and Fabian et al. have paved the way. It might be interesting if the authors provide readers opportunity to follow the origins of the active techniques and historical development. One additional paragraph in the introduction section could help. Technical advantages over the balloon observation systems from the US and Japan teams might be also interesting.

To acknowledge historic instrument development and balloon observations, the introductory sentence was extended with the following paragraph:

*"High-altitude balloons continue to be the only means for in-situ observations of chemical composition at altitudes that cannot be reached by aircraft, i. e. above ca. 20km. Lightweight instrumentation, such as for example ozone sondes, can be lifted with small weather balloons, that can be launched routinely. Many trace gases, however, can only be measured with more complex instruments or from sampled air analysed post-flight in the laboratory (Ehhalt 1980, Fabian 1981). Cryogenic air sampling is an established method for the efficient collection of air samples in the stratosphere (Ehhalt 1974, Lueb et al. 1975, Schmidt et al. 1984, 1987) to obtain observations of trace gas profiles from the stratosphere."*

Our measurement report does not provide advances in balloon observation systems per se but focuses on the comparison of established observation techniques for greenhouse gases that could also be deployed on other airborne platforms. Therefore, we refrain from comparing with balloon-based observational efforts in the US or Japan or other regions.

Age of air calculation should be elaborated so that readers can understand principles of the method better. Although I understand that this study stands on the past technical efforts, needed is more exact explanations about the method, not the software. The package should assume an age spectra formulation to find the best-match mean age of air. It could be more clearly explained why the surface reference time series of CH4 is used for calculation of the mean ages from CO2 and SF6 (were they or stratospheric data used for the CO2 correction?). What kind of assumptions brought the bias correction applied to SF6-derived ages of air might be also more explained with additional sentences, in addition to reference to Garny et al. paper. Although they are complex, the authors should provide essential parts. Since age of air is not uniquely determined from the data, I think the manuscript should provide the method in a way independently understandable.

The description of the age of air calculation was extended to make it understandable more easily without having to refer to the Garny et al. references. Please see response to specific comment on subsection 2.6 below for details.

Figures 2, 3, 4, 5 and 7 might be merged into a multi-panel graph. I am interested in comparing vertical profiles of different species particularly for correspondences of gradients among species. The tropopause could be marked, otherwise readers wonder at places like "in the troposphere." Profiles down to the troposphere are great advantage of this study.

The tropopause has been added to all profile figures as suggested (see below for further details). Merging all profile graphs into one figure creates fairly small panels. We have instead merged Figures 2 and 3, showing CO2 and CH4, and Figures 4 and 5, showing N2O and SF6, into one figure, respectively.

**Specific comments**

Throughout the manuscript: I've been wondering what HEMERA and TWIN stand for.

HEMERA is actually not an acronym, but the programme name refers to the ancient Greek goddess Hemera, the mythological personification of daylight.

The name TWIN of the gondola refers to the symmetric structure of the gondola frame. We have added this information to the text and to the caption of Figure 1. Although the name of the gondola may not be relevant for the discussion of the flight, but it helps to distinguish the payload from other HEMERA payloads launched in August 2021.

P2 L51: Though with some difference in methods, use of halocarbons for estimating mean age of air can be further back in time e.g., Daniel et al. (1996) and Harnisch et al. (1999). It might be worth mentioning that the idea existed much earlier than the recent extensive studies like Ray et al. (2017) and Leedham Elvidge et al. (2018).

In addition to the already cited work by Volk et al. (1997), we have added the suggested references in the revised version of the manuscript.

P3 L63: I could not find the companion paper (Laube et al. 2024) available. It must be made available before considering the decision of this manuscript. Otherwise this manuscript should provide more information without mentioning the reference.

Unfortunately, the processing of the companion manuscript had been significantly delayed. Laube et al. 2024 has thus changed to Laube et al. 2025, and it is now under review for the journal Atmospheric Measurement Techniques. In the meantime, the preprint has been published on EGUsphere:

> Laube, J. C., Schuck, T. J., Chen, H., Geldenhuys, M., van Heuven, S., Keber, T., Popa, M. E., Tuffnell, E., Vogel, B., Wagenhäuser, T., Zanchetta, A., and Engel, A.: Vertical distribution of halogenated trace gases in the summer Arctic stratosphere determined by two independent in situ methods, EGUsphere [preprint], https://doi.org/10.5194/egusphere-2024-4034, 2025.

The reference in the manuscript will be updated correspondingly.

P4 L84: "position data" do they include the GPS altitude? Regarding this, the altitude coordination in this study is unclear. Are the altitudes in Figure 2 and the following figures the GPS altitude recorded by the Pico-SDLA instrument? Or was pressure recorded and converted to altitude?

The position data include GPS altitude, latitude and longitude from the GNSS system onboard Pico-STRAT Bi Gaz. Pressure measurements were provided by Pico-STRAT Bi Gaz, that uses a ParoScientific Inc absolute gauge to measure air pressure at high accuracy (0.01%).

The information in the payload subsection is extended correspondingly in the revised version and now reads:

*"Position data as well as ambient pressure and temperature were recorded by the Pico-STRAT Bi Gaz instrument. The position data include GPS altitude, latitude and longitude from a GNSS system, pressure measurements are performed with a ParoScientific Inc. absolute gauge, and ambient temperature values are recorded by three fast-response temperature sensors (Sippican).''*

Axis labels were modified to "GPS altitude [km]".

P4 Section 2.2: It is regrettable that the Pico-STRAT Bi Gaz instrument for CO2 and H2O did not work well. Looking at Figure 2, I hope successful measurement at next opportunity as the present profiles were all from "reconstruction" from subsequent analyses.

The Pico-STRAT Bi Gaz instrument, which measures $CO_2$ and $H_2O$ mole fractions was affected by an external RF modulation, which distorted the $CO_2$ spectra. Water vapour measurements seem much less affected and the measurements are still valuable, but were not used.

Nevertheless, since the instrument was flying inside a larger gondola, water vapour measurements, are contaminated by the outgassing balloon wake from the cold point tropopause. The amplitude of this contamination slightly increases from 5 to 15 ppmv as the balloon ascent in the stratosphere, due to the increased number of outgassed water vapour molecules. The natural mean stratospheric water vapour mixing ratio is around 5 ppmv at high latitudes. In the troposphere, the natural mixing ratio dramatically increases by 4 to 5 orders of magnitude, which renders the contamination negligible.

This issue is well known and in dedicated campaigns, focusing on stratospheric water vapour studies, the hygrometers are located far away from the balloon (distances higher than 30 m) to reduce the contamination effect in the stratosphere. In the frame of this flight, measuring inside the gondola was done since the instruments were flown in the contaminated environment. Please note that the $CO_2$ profile from Fig.2 comes from AirCore measurements, while $CH_4$ profiles from Pico-STRAT are original profiles without any reconstruction.

P5 Table 1: I think that this table could be somehow merged with Table 3, because I have sometimes confused with which instrument for which species. A merged table might provide a quick look for such correspondences.

Merging Table 1 into Table 3 has turned out to result in a difficult to read table. We therefore repeated some information from Table 1 in Table 3 to make it easier to relate the given information to the text. Please note that following a suggestion by the second reviewer, an additional table, comparing technical details of the different AirCores, was included in the revised version of the manuscript.

P8 L192: Is there a reference that described usefulness of the cotton filter for COS measurements? Does co-existence of ozone and COS deteriorate their stability in the stainless steel flask? Such an explanation is valuable and desirably presented in this manuscript or the companion one.

Andreae et al. (1985), Hoffman et al. (1992), Persson and Leck (1994), Engel and Schmidt (1994) described the risk of possible biases due to reactions of ozone with reduced sulfur compounds and/or other tracers, both at tropospheric and stratospheric altitudes. Hoffman et al. (1992) and Persson and Leck (1994), in particular, focused on cotton scrubbers for tropospheric sampling. Engel and Schmidt (1994) observed depleted COS without ozone removal. More insights will be presented in Zanchetta et al. (in preparation, 2025).

All publications mentioned here are included in the references of the manuscript.

P8 Section 2.5: I think that it would be valuable to describe how much sample amount (volume) was used for each analysis. This information should be critical when the campaign was planned.

The Frankfurt GC-MS uses 1l per analysis, each sample was analysed twice. The QCLS runs for 5 minutes at 50sccm amounting to 250 ml. The CRDS uses 35ml/min over 180 s, adding up to 105ml, the GC-ECD setup used approximately 270 ml. The GC-MS analysis at FZJ used 250ml.

This information has been added to the manuscript upon revision extending the first paragraph of subsection 2.5. Air Sample Analysis:

*"All flasks of the cryo sampler were analysed at GUF using gas chromatography/mass spectrometry (GC-MS) for halocarbons, by gas chromatography/electron capture detection (GC-ECD) for CFC-12 and SF6, and with the continuous-flow CRDS used for AirCore analysis for CO2, CH4 and CO. The sample volume used for these analyses was two times 1 L for GC-MS, and approximately 0.27 L and 0.11 L for GC-ECD and CRDS analysis, respectively. The cryo sampler was then transferred to RUG where the continuous-flow QCLS used for AirCore analysis was deployed to measure CO2, CH4 and N2O from the sample flasks. This used approximately 0.25 L of sample volume. Last, samples were analysed for SF6 at FZJ using a GC-MS setup that needs 0.25 L of sample volume."*

P9 L219: How long does it take to have the instrument output stabilized with the flow rate ~50 sccm. I imagine that those stable data points were taken to calculate the measurement value and the time for stabilization determined the amount of sample used.

The following paragraph was added to the next to discuss the aspect of instrument stabilization for the QCLS:

*"The instrument stabilization is of 'double exponential' character, i. e., exhibiting an initially rapid approach to the final value, but then taking a long time to become stable. This depends on species and is more pronounced for CO2 than for CH4. The measurement duration of ~5 minutes, chosen to conserve precious sample, does not for all samples result in quantitative replacement of the previous sample. In order to maximize the value of our analysis, a double exponential function was fit to the measurement data and used to predict the true sample value, i.e., the asymptote of the function. The deviations between the asymptote and the mean of the last 60 seconds of a sample was for almost all samples smaller than the uncertainty of either. That means that this method does not make unjustified assumptions and does not add significant uncertainty to the results."*

P10 Section 2.6: As in my earlier comment, I hope this section be reformulated so that readers can correctly follow principles, assumptions and processes to calculate the age of air.

The section has been extended for the revised manuscript as follows:

*"**2.6 Age of air calculations**

The mean transit time that it took for all contributions to an observed air parcel to arrive at the observation location from their respective entry points into the stratosphere is defined as the mean age of air. Thus, the observed mixing ratio of an inert trace gas at some place in the stratosphere is determined through the distribution of transit times, called the age spectrum, and the long-term change in its mixing ratio at the entry point. Commonly, the age spectrum is mathematically described as an inverse Gaussian function, and the mean age of air is the first moment of this distribution (Hall and Plumb, 1994; Waugh and Hall, 2002). Mean age of air values were derived from $SF_6$ measurements and independently from simultaneous $CO_2$ and $CH_4$ measurements following the procedure described by Garny et al. (2024b) using the AoA_from_convolution python package version 1.0.0 (Wagenhäuser et al., 2024). In brief, this method calculates the expected mixing ratios of an inert trace gas through a mathematical convolution of the age spectra for different mean age values and the mixing ratios time series at the entry point. The mean age is then determined from the best match between the observations and the mixing ratios resulting from this forward calculation. The derived mean age is the age value for which the convolution best fits the observed mixing ratio.*

*To calculate mean age from tracer observations, also the time series of the respective tracer at a reference surface is needed. For this purpose, the AoA_from_convolution package uses the NOAA Greenhouse Gas Marine Boundary Layer Reference for $SF_6$, $CO_2$ and $CH_4$ trace gas mixing ratio time series at the tropical surface ±17.5° around the equator (Lan et al., 2021; Garny et al., 2024c). The inverse Gaussian describing the age spectrum is parameterised assuming a ratio of moments as described in Garny et al. (2024b). Mean age values below 1 year are omitted due to numerical reasons of the software implementation. Regarding $SF_6$, these mean age alculations do not account for the mesospheric sink, which leads to apparently older $SF_6$ mean ages (Leedham Elvidge et al., 2018; Garny et al., 2024a).*

*For $CO_2$, the software first uses $CH_4$ mixing ratios to account for stratospheric $CO_2$ production from $CH_4$ degradation. This corrected $CO_2$ mixing ratio is then used to derive the mean age. Note that the seasonal cycle of $CO_2$ propagates into the lower stratosphere, and it is impossible to disentangle the seasonal cycle from the long-term increase. Therefore, mean age values below 2 years derived from $CO_2$ measurements are problematic (Garny et al., 2024a)."*

P10 L265: "in the troposphere" the tropopause should be presented. Otherwise readers cannot be clear about which altitude range is mentioned.

We agree that adding the tropopause helps to read the graphs and have added it to all profile plots. We have derived the tropopause height from the radiosonde data during the balloon ascent by calculating the slope of the temperature profile. The tropopause height

has been assigned at 10.5km where the temperature change with altitude first reached the value of 2K/km following the WMO definition. A corresponding explanation has been added to the revised manuscript.

 It is interesting to see the ups and downs in CH4 above 20 km observed by the Pico instrument. Is there any possible mechanism that could shape such layers?

The origin of such structures is under investigation, however, we do not have an explanatory hypothesis and believe that this is beyond the scope of a measurement report. Some of the features are associated with ozone structures, observed by an ozone sonde flown 2 hours later than the ZPB flight of TWIN which is analyzed here. Similar structures were also observed during an earlier flight of the Pico-SDLA instrument from Kiruna in 2016.

These structures were also resolved with the sub-sampled MegaAirCore and this is discussed in more detail in the companion paper by Laube et al. 2025.

**Technical Comments**

 "known" to "considered"

Changed as suggested.

 "caused"

 "Aircore" to "AirCore"

 "altitde" to "altitude"

 "Age of Air" to "age of air"

These typos have been corrected in the revised version of the manuscript.

---

## Author Comment (AC2)

**Schuck et al., Measurement report: Greenhouse gas profiles and age of air from the 2021 HEMERA-TWIN balloon launch**

**Response to comments by Anonymous Referee #2**

We thank the anonymous reviewer for their comments and for their suggestions to improve the manuscript. All comments are addressed in the following with the reviewer's comments printed in blue, and the responses in black.

**General Comments:**

The manuscript outlines measurements made during the August 2021 large-balloon-assisted launch of the HEMERA-TWIN gondola, which hosted discrete flask sampling systems, AirCore samplers and continuous in situ instrumentation for measurement of long-lived atmospheric trace gases. Profile data are compared to flask sampler data, and stratospheric mean age of air is calculated from flask and AirCore samples of CO2 and SF6.

Overall, this measurement report provides updated large-balloon observations from Kiruna and new and important observation-based calculations of stratospheric mean age of air worthwhile of publication. Some scientific, technical and organizational comments below are recommended to be addressed before publication, however (see specific comments below).

- There are several references to measurements made, but not used in this manuscript and/or described elsewhere (Laube et al., 2024 – not currently accessible). Text throughout the manuscript could be tightened up a bit to not cause undue confusion.

Unfortunately, the processing of the companion manuscript by Laube et al. was delayed. It is now accessible as

*Laube, J. C., Schuck, T. J., Chen, H., Geldenhuys, M., van Heuven, S., Keber, T., Popa, M. E., Tuffnell, E., Vogel, B., Wagenhäuser, T., Zanchetta, A., and Engel, A.: Vertical distribution of halogenated trace gases in the summer Arctic stratosphere determined by two independent in situ methods, EGUsphere [preprint], https://doi.org/10.5194/egusphere-2024-4034, 2025.*

Although some measurements are not discussed in the measurement report, we still consider it relevant to mention them for completeness. In addition to the Laube et al. manuscript, there will be a further publication by Zanchetta et al. later in 2025.

- More details are needed in the onboard sampling systems and analytical descriptions, in addition to the mean age calculation in order for methods to be sufficiently understandable and repeatable by readers.

For the revised version of the manuscript, we have significantly extended the description of the age of air calculation (see also response to specific comment below). Regarding the sampling systems and analytical methods, we think the description should remain concise and should not repeat too many details that have been discussed in depth in dedicated measurement technique publications.

- The visualization and description of the profiles in each figure could be improved upon, and separate AirCore profile data figures could be merged into one multi-panel figure for comparisons of species' profiles and their corresponding description within the text.

Merging all profiles into one multi-panel figure results in fairly small panels. We have instead merged Figures 2 and 3, showing CO2 and CH4, and Figures 4 and 5, showing N2O and SF6, into one figure, respectively.

**Specific Comments:**

Figure 1: I am wondering where each package was located on the gondola. Could some packages in this figure be somehow labeled to indicate where measurements were made?

Labels were added to the figure during manuscript revision.

Table 1:

- Five – L82? (or three – L118?) different AirCores were used here, but it would be nice to have these (with perhaps IDs for each AirCore and dimensions, configuration, dryer, no dryer, etc.) described in Table 1 explicitly. Please also add ambient pressure and temperature sensors to Table 1 as these are equally important measurements.

RUG:

| Flight date and takeoff time (UTC) | Location and flight code | Coordinates and apogee | Instrument | Instrument features | Inlet features |
|---|---|---|---|---|---|
| 12/08/2021 21:18 | Kiruna KRN | 67°53' N 21°04' E 33.1 km | AirCore | 36 m x Ø 3/16" + 38 m x Ø 1/8" V ~ 829 cm³ | Free inlet |
| 12/08/2021 21:18 | Kiruna KRN | 67°53' N 21°04' E 33.1 km | AirCore | 37 m x Ø 3/16" + 39 m x Ø 1/8" V ~ 859 cm³ | Dryer (Mg(ClO4)2) |

The Instrument package included five AirCores of which three are discussed in the text. The reason for the exclusion of the second GUF operated AirCore from further data analysis is discussed in section 2.3 of the manuscript (because of a failure of the CO spiking experiment). For the fifth Aircore we refer to Laube et al 2025, which is now accessible on EGUsphere.

Regarding the suggested extension of Table 1, we think that it would double too much information. We only discuss one individual flight, thus this information does not need to be repeated in a table several times. In addition, such a table would compare very different types of instrumentation. The instrument features are therefore not comparable and they are not well suited to distinguish the instrumentation.

Therefore we add an additional table to subsection 2.3. only for the AirCores used.

- LISA and the FZJ AirCores are included in this table, but data are not presented. This is contrary to the approach taken in Table 3. Pico-STRAT Bi GAz is also not added in Table 1. I would suggest either adding all, but with "*" for measurement artifacts and/or descriptions in other, cited papers, or remove otherwise.

We have corrected Table 1 for the consistent mentioning of Pico-STRAT Bi GAz. In addition, we have extended the labelling of Table 3 to make it more clear what is included. LISA and FZJ AirCore are not included in Table 3, but still we think for completeness the full payload should be listed in Table 1.

L125: Not all AirCores (e.g. FZJ) were analyzed by continuous-flow analyzers. Please correct/clarify. Were similar fill gas mixtures used for each AirCore?

We have modified Table 3 to make it more clear which AirCore was analyzed with which analyzer.

The fill gas mixtures were different for GUF and RUG, but from each institute both AirCores were prepared with identical fill gas mixtures. This has been made more clear in the text:

*"The initial fill gas of both GUF AirCores had CH4 and CO2 mixing ratios close to those expected in the middle stratosphere but was spiked with CO, resulting in a CO mixing ratio of 1436.41 ppb. Mixing with the remaining fill gas is taken into account during the retrieval as described by Wagenhäuser et al. (2021). Thus, the uppermost part of the AirCore profile can be used for scientific evaluation as well."*

L139/L159: It seems that the GUF and RUG AirCores samples were processed somewhat differently, from fill/push gas mixing corrections to the altitude attribution of the samples.

Please describe in more detail what the differences are between both GUF and RUG methods so that this can be assessed better.

The following clarification, summarizing the details given in subsection 2.3 was added to the revised manuscript:

*"The main difference in the processing procedures from GUF and RUG is the fill gas correction. For the RUG AirCores, no fill gas correction is performed, therefore profiles do not extend as high as GUF profiles. An additional uncertainty is introduced for RUG AirCores because of the failure of the automatic closing of the inlet after touch down. This may have cause a loss of sample that could not be corrected."*

A more detailed comparison of altitude attribution would certainly be a publication on its own and would require launching several AirCores multiple times and an exchange of not only calibration standards but ideally also analyzers and data processing code between laboratories. This has been attempted recently within the OSTRICH project for which data evaluation is in progress.

Section 2.3: You might consider a 2.3.1 'GUF AirCores' vs. 2.3.2 'RUG AirCores' for organization

There is a lot of overlap between these two and such a seperation would become very repetitive. We therefore opted to not split Section 2.3. into to subsections.

 There is no mention of an ambient water vapor measurement onboard the gondola. It is difficult to examine water effects on the profiles (either dried or undried) without this type of measurement. For example, if the atmosphere is dry, no effects will be seen, so some clarification here and/or below in the description of differences seen between these profiles should mention something about the average tropospheric water vapor content above Kiruna.

Water vapour may cause dilution during analysis, as well as biases due to dissolution of some gas species in it. Although not directly measured, it seems that its effects on the measured gas species would be marginal, if not absent. The major differences are observed on the stratospheric part of the profile, where water vapour is scarce and can be therefore excluded from the possible reasons that could explain this bias.

Unfortunately, the second PICO spectrometer did not deliver high quality data due interferences from the gondola (RF coupling). This affected mainly $CO_2$ and although water vapour measurements seem less affected, we preferred not to use them.

Please define $Mg(CLO_4)_2$.

The name 'magnesium perchlorate' was added on first mentioning of the chemical formula of the drying agent.

Please also describe in more detail the altitude attribution used here. Is this similar to Tans, 2022 or Wagenhauser et al., 2021? How do these methods compare/differ?

For GUF AirCores, altitude attribution follows Wagenhäuser et al. 2021:

*"Altitude attribution was performed as described by (Engel et al. (2017) and Wagenhäuser et al. (2021). The start and end points of the AirCore sample in the measurement time series were determined using the known mixing ratios of the remaining fill gas that the AirCore tube is filled with prior to the launch and the push gas that is used to push the sample air towards the Picarro instrument during the post-flight analysis. Prior to launch, the AirCore is filled with a fill gas and closed at one end. During ascent, the fill gas empties into the atmosphere with decreasing ambient pressure. At the ceiling height, a small amount of fill gas remains in in the AirCore tube and during descent is pushed towards the closed end of the AirCore. During analysis, which is performed in the reverse direction, the remaining fill gas marks the start of the AirCore sample in the measurement time series. In this procedure an easily distinguishable fill gas facilitates the analysis: for the HEMERA 2021 launch, we used a gas with CH4 and CO2 mixing ratios close to those expected in the middle stratosphere and spiked with CO, resulting in a CO mixing ratio of 1436.4 ppb."*

For RUG AirCores, the appoarch describe by Membrive et al. (2017) is used:

*"The altitude attribution follows the pressure equilibrium approach described by Membrive et al (2017). Knowing pressure and temperature, it is possible to calculate the number of sampled moles for each sampling altitude. Each number of moles during sampling can then be associated to the number of moles of each step of the gas analysis, allowing the altitude mapping retrieval of the sample."*

Table 3:

- The text mentions that this table describes precision for both flask and AirCore measurements (L240), so the caption should mention this.

Upon manuscript revision, we have extended the caption to *"Instrumental precision and average error of analyses of flask samples and AirCores."*

- It is a bit confusing with some species mentioned in this table. COS is mentioned for the RUG QCLS, but Table 3 precisions and results are not described. FZJ analysis precision by GC-MS is stated in Table 3 for only SF6 results with other species presented elsewhere.

Regarding Table 3, we restricted the information to only those species discussed in the manuscript. COS precision on the QCLS is 20 ppt. Results should be presented in Zanchetta et al. (2025), in preparation.

Please see comment above for Table 1.

While Table 1 is meant to give an overview of the full payload for completeness, we do not think that adding more information to Table 3 is informative. Additional numbers in the table would also require additional information on further instrumentation in the main text which would not be relevant for the discussion of the greenhouse gas profiles. We therefore restrict Table 3 to those species actually discussed in the manuscript, although this is contrary to the approach taken for Table 1.

Section 2.5: Could you please clarify the progression of air sample analysis and how many flasks were analyzed by each organization? Are all flask samples analyzed for halogenated species at GUF by GC-MS, by GC-ECD for CFC-12 and SF6, and by CRDS for CO2, CH4 and CO and then subsequently transferred to RUG and FZJ for additional species analysis?

The sample analyses were indeed carried out that way. We have added this information as an introductory paragraph to subsection 2.5. of the revised manuscript:

*"All flasks of the cryo sampler were analysed at GUF using gas chromatography/mass spectrometry (GC-MS) for halocarbons, by gas chromatography/electron capture detection (GC-ECD) for CFC-12 and SF6, and with the continuous-flow CRDS used for AirCore analysis for CO2, CH4 and CO. The sample volume used for these analyses was two times 1 L for GC-MS, and approximately 0.27 L and 0.11 L for GC-ECD and CRDS analysis, respectively. The cryo sampler was then transferred to RUG where the continuous-flow QCLS used for AirCore analysis was deployed to measure CO2, CH4 and N2O from the sample flasks. This used approximately 0.25 L of sample volume. Last, samples were analysed for SF6 at FZJ using a GC-MS setup that needs 0.25 L of sample volume."*

Section 2.6: Please describe in a bit more detail how AoA_from_convolution works for the reader within the text. It might also be clarified in subsequent text that mean ages are calculated and used.

The section has been extended for the revised manuscript as follows:

*"2.6 Age of air calculations*

*The mean transit time that it took for all contributions to an observed air parcel to arrive at the observation location from their respective entry points into the stratosphere is defined as the mean age of air. Thus, the*

*observed mixing ratio of an inert trace gas at some place in the stratosphere is determined through the distribution of transit times, called the age spectrum, and the long-term change in its mixing ratio at the entry point. Commonly, the age spectrum is mathematically described as an inverse Gaussian function, and the mean age of air is the first moment of this distribution (Hall and Plumb, 1994; Waugh and Hall, 2002). Mean age of air values were derived from $SF_6$ measurements and independently from simultaneous $CO_2$ and $CH_4$ measurements following the procedure described by Garny et al. (2024b) using the AoA_from_convolution python package version 1.0.0 (Wagenhäuser et al., 2024). In brief, this method calculates the expected mixing ratios of an inert trace gas through a mathematical convolution of the age spectra for different mean age values and the mixing ratios time series at the entry point. The mean age is then determined from the best match between the observations and the mixing ratios resulting from this forward calculation. The derived mean age is the age value for which the convolution best fits the observed mixing ratio.*

*To calculate mean age from tracer observations, also the time series of the respective tracer at a reference surface is needed. For this purpose, the AoA_from_convolution package uses the NOAA Greenhouse Gas Marine Boundary Layer Reference for $SF_6$, $CO_2$ and $CH_4$ trace gas mixing ratio time series at the tropical surface ±17.5° around the equator (Lan et al., 2021; Garny et al., 2024c). The inverse Gaussian describing the age spectrum is parameterised assuming a ratio of moments as described in Garny et al. (2024b). Mean age values below 1 year are omitted due to numerical reasons of the software implementation. Regarding $SF_6$, these mean age alculations do not account for the mesospheric sink, which leads to apparently older $SF_6$ mean ages (Leedham Elvidge et al., 2018; Garny et al., 2024a).*

*For $CO_2$, the software first uses $CH_4$ mixing ratios to account for stratospheric $CO_2$ production from $CH_4$ degradation. This corrected $CO_2$ mixing ratio is then used to derive the mean age. Note that the seasonal cycle of $CO_2$ propagates into the lower stratosphere, and it is impossible to disentangle the seasonal cycle from the long-term increase. Therefore, mean age values below 2 years derived from $CO_2$ measurements are problematic (Garny et al., 2024a)."*

L271: Combining CO2, CH4, N2O profiles in a single figure would be advantageous to your description of the profiles within the text, and would allow the reader to better compare/contrast the variability between profiles.

We have merged CO2 and CH4 into one figure, but, to avoid too narrow panels, have combined the N2O profile with the SF6 profiles.

Figures 2-6:

- It is very difficult to distinguish the colors between RUG AirCore 'wet' and 'dry'.

For the revised figure, the colour of the dry RUG AirCore profile was darkened and the linestyle was made different in addition. Also linewidth was enhanced to make the lines better visible.

- Cryosample and other markers overlap each other at times and it's difficult to see what's underneath of these. Consider decreasing marker size here.

It is difficult to avoid an overlap of markers, as results are often close, and still keep a decent marker size to make the markers distinguishable. Marker size was reduced for the revised figures, and markers for GUF and RUG cryosampler results have been set to different sizes to decrease the overlap.

- Line width on PICO data covers AirCore profile data as well; consider reformatting so that all profiles can be sufficiently visualized.

During revision of the figures the plotting order and line width were modified to better visualize the AirCore profiles and PICO data.

L274: The altitude registration offset looks to be greater than 300 m (maybe 500-800m?), hence my request for additional description of the altitude attribution techniques above and potentially additional explanation for this offset in L281.

This is mentioned in subsection 2.3:

*"The vertical resolution of the GUF AirCores ranges from about 1000 m at 25 km to better than 300 m around the tropopause and in the troposphere."*

We agree with the reviewer that 300 m is an average value for the vertical offset and at different parts of the profiles is larger. The statement was changed to listing the range of altitude deviation of 200-700 m

L303: The AirCore averaging kernel or diffusion is stated multiple times throughout the text. Could you include calculated uncertainties for this either in the text or in the figures above to indicate how much of an uncertainty the AirCore storage diffusion has?

Again, we refer to subsection 2.3:

*"The vertical resolution of the GUF AirCores ranges from about 1000 m at 25 km to better than 300 m around the tropopause and in the troposphere."*

At lower altitudes, the effect of molecular diffusion on the vertical resolution is larger, because of the larger tube diameter for the lower part of the profile. At the top of the profile, mixing in the analyzer cell during post-flight analysis is the dominating effect.

This information has been added to the text in the revised manuscript:

*"The vertical resolution of the GUF AirCores ranges from about 1000 m at 25 km to better than 300 m around the tropopause and in the troposphere. However, the geometry of the AirCore plays a central role in this uncertainty, so the three individual sections of the GUF AirCore with their different internal diameters and lengths must be taken into account. At lower altitudes, the effect of molecular diffusion on the vertical resolution is larger because of the wider tube diameter. At the top of the profile, mixing in the analyzer cell during post-flight analysis is the dominating effect. Further details of the AirCore data analysis including the altitude attribution and fill gas correction were reported by Wagenhäuser et al. (2021)."*

L314: Can you offer any thoughts as to why AirCore CO2 (and N2O) seems to be lower than the cryosampler data of the same species? Could it be a problem with one or either of these sampling methods?

Because of the good agreement of the independent analyses at GUF and RUG for CO2, we believe that is more likely a sampling effect than measurement related. Although AirCore profiles were averaged over the altitude range sampled by the cryo sampler, this may not be fully accurate, as AirCores themselves have an averaging kernel. Thus, the averaging might be over slights different altitude ranges. The deviations seem largest at altitude regimes where the observed vertical gradient is steep. Therefore, small sampling artifacts or uncertainties in the altitude attribution would likely have large effects. In the comparison of

methane with the fast PICO data we also see some deviations, which are largest in the vicinity of the observed layered fine structures recorded by PICO. At higher altitudes, AirCore fill gas correction might also be relevant or impurities of the tubing and its inert coating at the positions were the individual tubing pieces of the AirCores are connected. For this latter effect, $CO_2$ is known to be more sensitive to than other species.

**Technical Comments:**

L17: 20 km 'MSL'?

For the altitude regime discussed here, referring to sea level seems not absolutely needed and we do not add m.a.s.l. or similar for better readability.

L21: It is perhaps important to note that balloon data are also needed to validate and "calibrate" remote-sensing instrumentation like those in the TCCON and NDACC, not only to supplement them.

We have added this aspect to the text as follows:

*"For substances which are measurable with remote-sensing methods, data from balloon-borne air samples can also be used for satellite retrieval validation or to validate ground-based measurement networks such as the Total Carbon Column Observing Network (TCCON) and the Network for the Detection of Atmospheric Composition Change (NDACC) deploying Fourier transform infrared spectrometers (Zhou et al. 2018)."*

We do deliberately not add the term "calibration" as this means something fairly different for the types of measurement presented later in the manuscript.

L24: What type of balloon platform (large or small-volume balloon) and instrument (flask, in situ) was used here? Please describe, as AirCore could potentially be used as a low-cost option that refutes the statement in L25.

Earlier launches of the cryo sampler also used for the HEMERA TWIN flight were evaluated for the Kolonjari et al. study. This information was added to the revised manuscript:

*"This was for example recently demonstrated for vertical profiles of HCFC-22 (Chlorodifluoromethane, CHClF2) measured by the ACE-FTS satellite instrument (Kolonjari et al. 2024) using data from the analysis of flasks sampled at ground-based sites, on board aircraft and sampled during four earlier flights of the identical cryo sampler used for the HEMERA TWIN launch."*

L29: Please define chemical species names of CO2 and CH4, but also elsewhere in the manuscript for completeness.

Chemical species names are defined on the respective first occurrence in the text. For $CO_2$ and $CH_4$ this is in Line 6. All other chemical species were double checked. For magnesium perchlorate, the missing definition was added to the chemical formula.

L33: The AirCore approach, providing higher resolution profiles and additional air sample measurements, seems like an improved method over discrete flask sampling methods. Consider replacing the word "complementary"

Regarding CO2 and CH4, AirCores indeed are an improved method. However, since they provide a profile that due to diffusion undergoes a different averaging compared to flask samples, it seems preferable to call it complementary. Flask samples still offer advantages over AirCores, such as the larger number of species that can be analyzed or the possibility of repetitive analysis for more reliable results and the option to store samples for a longer period to perform analyses with instrumentation that is slow or may not be available right after a flight. We therefore prefer to remain with the initial wording of "complementary". Subsampling of AirCores as one option to overcome these disadvantages of AirCores is demonstrated in the Laube et al companion paper.

L39: What does HEMERA-TWIN stand for, if anything?

TWIN refers to the symmetric shape of the gondola frame. This has been added to the text. We use the naming HEMERA-TWIN to make this launch distinguishable from other balloon launches also performed during the mission in August 2021 from Kiruna.

L40: What are 'reference methods'? It might be more accurate to say that the gondola instrumentation allowed simultaneous comparison of several different measurement methods and not to say that one is more accurate than the other.

In the revised manuscript, 'reference methods' is replaced with 'other measurement methods'. In the context of trace gas measurements, 'reference methods' is a term used in EU air quality legislation and it this may be confusing, as we did not mean to refer to any specific measurement technique here.

L42: Please describe why calculation of the mean age of air is significant and useful in a bit more detail for context.

To motivate the calculation of age of air values, the introduction has been extended. The respective paragraph of the revised manuscript reads:

*"To investigate stratospheric transport time scales, the concept of mean age of air has proven to be a useful tool. As the stratospheric circulation cannot be observed directly, a quantity that can be derived from observations of trace gas mixing ratios is needed to characterize stratospheric transport. Mean age of air can be used to diagnose the current overall strength of the large-scale Brewer-Dobson, but also allows to investigate changes that might be a consequence of climate change (Hall and Plumb, 1994; Waugh and Hall, 2002; Engel et al., 2009, Garny et al., 2024b)."*

L49: 'monotonic'

'Monotonous' was kept, and we will clarify the correct wording during the final typesetting and editing process.

L59: ..bag sampling, [and] air sampling by means of …

Changed to *"(...) in-situ spectrometric analysis, cryogenic air sample collection in stainless steel canisters, bag sampling, and air sampling by means of AirCores."*

L75: an average 'ascent' rate … and 31.2km 'MSL'?

As the sentence starts in the previous line with "Ascent" we prefer "altitude rate" here.

L82: Please define BONBON acronym

BONBON is the historic name of the cryo sampler device, but it actually is not an acronym. We have removed the name from the text.

L86: Please define LISA

The use of upper lower case of the line was modified to make this more clear: *"(...) LIghtweight Stratospheric Air sampler (LISA) (...) "*

L94: In Pico-STRAT Bi GAz, what does STRAT, Bi and GAz signify?

The Pico-STRAT Bi Gaz instrument is one of the instruments within the Pico-SDLA suite of instruments. It is a dual gas spectrometer ("Bi Gaz": dual gas in French) which has been adapted for long duration observations in the equatorial tropopause. The STRAT portion then refers to the Strateole 2 mission (long duration) for which the original Pico-SDLA has been adapted. This is most probably of a high significance to detail in the main manuscript.

We have modified the first paragraph on the Pico subsection as follows:

*"Pico is a balloon-borne spectrometer developed to probe vertical profiles of atmospheric $CH_4$ and $CO_2$ (Ghysels et al., 2011, 2014). During the HEMERA-TWIN flight, two Pico instruments were launched: Pico-SDLA $CH_4$ and Pico-STRAT Bi Gaz ($H_2O/CO_2$). The Pico-STRAT Bi Gaz ("Bi Gaz": dual gas in French) spectrometer is an evolution of the former Pico-SDLA instruments to suite long duration observations This adaptation has been initiated within the STRATeole 2 balloon-borne project. (Carbone et al., 2024). The Pico-SDLA $CH_4$ instrument performed well, whereas Pico-STRAT Bi Gaz measurements suffered from undesired electromagnetic interference for which the source remains undetermined, resulting in spectrum deformations for $CO_2$. Therefore, $CO_2$ measurements are unusable for this flight."*

L133: Here and throughout the manuscript, text switches from spelling out 'Frankfurt', or 'University of Frankfurt' and 'Groningen' vs. using the acronyms for each organization. Consider using just acronyms for simplicity.

During manuscript revision the acronyms GUF/RUG has been used after first introduction throughout the text replacing "University of Frankfurt / Groningen" as well as FZJ for "Forschungszentrum Jülich".

L149: part of the AirCore [CO2 and CH4?] profiles can be used…

'profile' changed to 'profiles' as suggested. The statement is applicable to all trace gases, depending on the analyzer used, therefore we do not explicitly mention CO2 or CH4 here.

L153: these are reported as dry mole fractions

In general we use 'mixing ratios throughout the manuscript. However, there were two inconsistent usages of 'mole fractions' instead which we changed to 'mixing ratios' as well.

L171: in subsection [space] 2.5 (Vincović et al. 2022; Tong et al, 2023).

[space] added.

L174: ..and [are] visible?

Changed as suggested.

L185: upper and lower parts, or just upper parts?

Changed to *'the upper and lower parts of the RUG profiles are not reported.'*

L290: Above states that data is only collected upon balloon descent, so please correct.

This line clarifies that for the PICO-SDLA $CH_4$ instrument this is not the case, as it measured during ascent and descent.

L293: Can you please state over how long of a sample collection time this averaging takes place?

The sample collections times vary from 43 s to 731 s. This information has been added to that line and to subsection 2.4

L308: The fine structure here is not seen in AirCores due to the diffusion of the samples, presumably.

This is most likely the reason. As AirCores by default apply some kind of averaging kernel to the profile, in situ spectrometry has advantages if such fine structures are to be resolved. This demonstrates the usefulness of the dataset from this balloon launch. Because of the heavy payload that could be launched, we are able to compare very different experimental approaches.

L309: Delete "University of Frankfurt,", "by", and add on average, [by]

Because there are two $CH_4$ measurements of the cryo sampler, "University of Frankfurt" is needed here.

Second part is changed to "on average by 35 ppb" during revision.

L326: age of air 'values'

Changed as suggested.

L335: value derived 'from'

Typo corrected as suggested.

Figure 7: The x-axis label should be years, correct? Caption: "..markers represent 'Age' of air"

The caption typo was corrected to "Age", for the x-axis we prefer to keep "Age of air" rather than years, as age of air is the derived quantity, whereas years is the unit it is reported in.

L372: Does the NOAA product come with a citation reference?

Supplementary to Garny et al. 2024 and the code repository, the reference to the NOAA MBL data has been added during revision:

*Lan, X., Tans, P., Thoning, K., & NOAA Global Monitoring Laboratory. (2023). NOAA Greenhouse Gas Marine Boundary Layer Reference - CO2. [Data set]. NOAA GML. https://doi.org/10.15138/DVNP-F961*